# Physiography, foraging mobility, and the first peopling of Sahul

Tristan Salles [1] ✉, Renaud Joannes-Boyau [2], Ian Moffat [2,3], Laurent Husson [4] & Manon Lorcery[1,4]

The route and speed of migration into Sahul by Homo sapiens remain a major research question in archaeology. Here, we introduce an approach which models the impact of the physical environment on human mobility by combining time-evolving landscapes with Lévy walk foraging patterns, this latter accounting for a combination of short-distance steps and occasional longer moves that hunter-gatherers likely utilised for efficient exploration of new environments. Our results suggest a wave of dispersal radiating across Sahul following riverine corridors and coastlines. Estimated migration speeds, based on archaeological sites and predicted travelled distances, fall within previously reported range from Sahul and other regions. From our mechanistic movement simulations, we then analyse the likelihood of archaeological sites and highlight areas in Australia that hold archaeological potential. Our approach complements existing methods and provides interesting perspectives on the Pleistocene archaeology of Sahul that could be applied to other regions around the world.

Our understanding of how anatomically modern humans interacted with the biotic and abiotic environments during their dispersal remains limited and mostly conceptual. This deficiency is particularly important when evaluating the peopling of Sahul (i.e., the single landmass including Australia and New Guinea which existed for most of the late Quaternary – Fig. 1a) prior to 65,000 years ago[1] and possibly as early as 75,000 years ago according to modelling estimates[2]. This peopling is notable for both the need to undertake a significant open water crossing from south-east Asian and the rapid post-arrival dispersal of people throughout the continent[1]. Previous studies of this extraordinary feat of exploration have either favoured dispersal strategies in multiple directions simultaneously or constrained routes along coastlines[3,4]. Between those end-members, a series of most-likely dispersal scenarios and preferential pathways[2,5–9] have been proposed and refined by incorporating the compounding influences of biogeographic, climatic, ecological, ethnographic, demographic and technological factors[6,9]. Initial approaches often relied on interpolations between dated archaeological sites while quantitatively or qualitatively

accounting for first-order environmental complexity. As an alternative, agent-based least cost path models[6,10,11] are convenient theoretical methods to predict the optimal migration routes of hominins through a landscape. They combine ecological niche analysis while taking into consideration group of travellers local perspective and incomplete knowledge. Other techniques, based on recent advances in genome sequencing and analysis[12], have provided insights into the dispersal and expansion of modern and ancient people but are constrained by both data availability and gene temporal expression patterns. To alleviate some of these limitations, new approaches have been proposed that combine ecomorphological agent-based modelling and machine learning[13], integrate time-varying climate conditions with advanced stochastic-ecological models[8,14], account for time-varying climate conditions with numerical reaction/diffusion human dispersal methods[15,16], or design pedestrian-transportation networks with visual-prominence aggregated maps[2,9].

In this study, we present an approach for understanding human hunter-gatherers' migrations based on Lévy walk foraging patterns.

[1]School of Geosciences, The University of Sydney, Sydney, NSW, Australia. [2]Geoarchaeology and Archaeometry Research Group, Southern Cross University, Lismore, NSW, Australia. [3]Archaeology, College of Humanities, Arts and Social Sciences, Flinders University, Adelaide, SA, Australia. [4]ISTerre, CNRS, Université Grenoble-Alpes, Grenoble, France. ✉e-mail: tristan.salles@sydney.edu.au

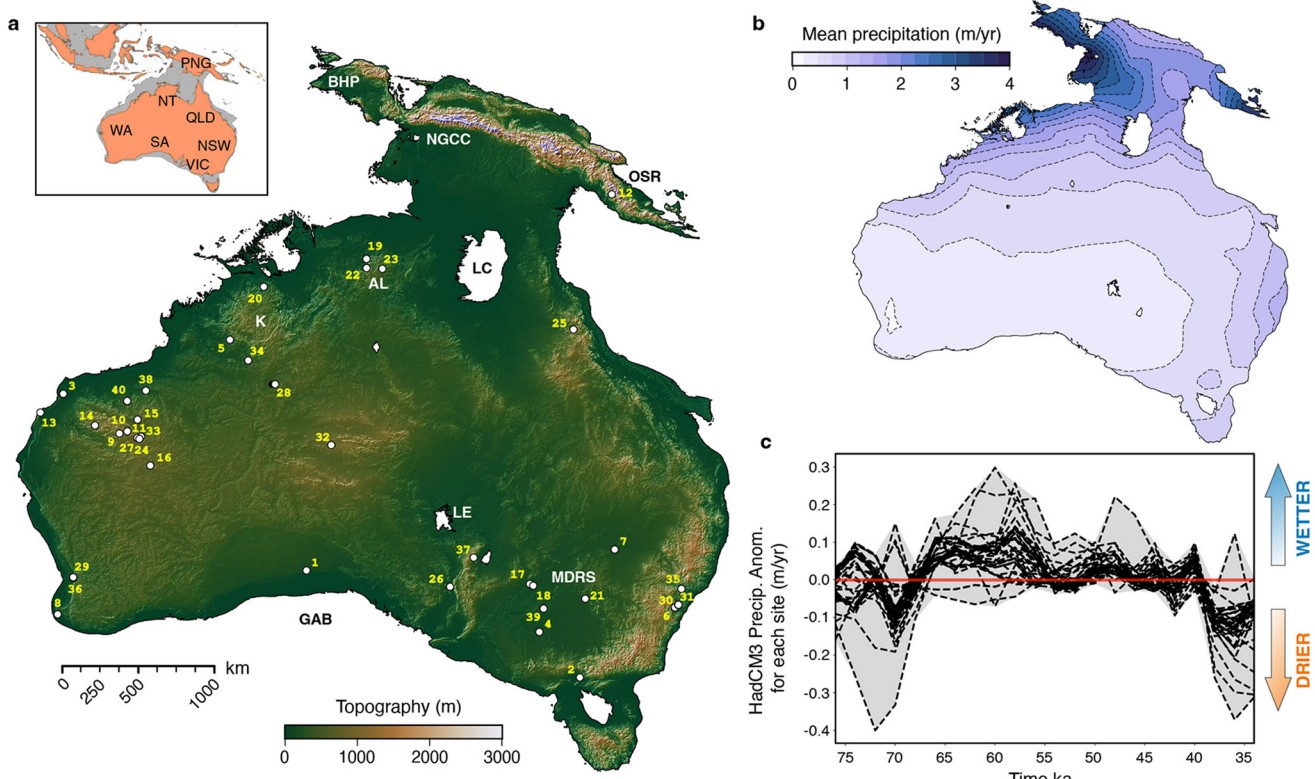

**Fig. 1 | Input data layers used to run the physiography model. a** Sahul topography around 65 ka obtained from regional and global-scale digital elevation models[9] assuming a coastline at −85 m isobath. The inset map shows modern coastlines superimposed on the coastline of Sahul during the approximate time of first human arrival (PNG: Papua New Guinea, NT: Northern Territory, WA: Western Australia, QLD: Queensland, SA: South Australia, NSW: New South Wales, VIC: Victoria). Archaeological sites older than 35 ka[9] with their site ID (defined in Supplementary Tab. 1 – BHP: Bird's Head Peninsula, NGCC: New Guinea Central Cordillera, OSR: Owen Stanley Range, LC: Lake Carpentaria, AL: Arnhem Land, K: Kimberley,

LE: Lake Eyre, MDRS: Murray-Darling River system, GAB: Great Australian Bight). **b** Sahul precipitation distribution from 75 to 35 ka and averaged from the multiple time slices (available every 2 kyr) based on the HadCM3 climate model[36]. **c** Distribution of precipitation anomalies for each archaeological site (Supplementary Tab. 1) over the simulated time interval. All data are available under Creative Commons Attribution 4.0 International Licence (http://creativecommons.org/licenses/). The open-source python interface for the Generic Mapping Tools (https://www.pygmt.org) is used for visualisation.

Lévy walks are a type of random walk search that consist of a combination of frequent short travelled distance steps (separated by periods of non-mobility or changes in direction) and rarer longer moves[17]. This type of displacement has been observed in many animal movement patterns[18], but also for modern hunter-gatherer populations in Tanzania, Botswana or Namibia[19–21], when searching for heterogeneously distributed food. While these latter studies[19,21] have focused on areas within regions already occupied by other human groups, Lévy walk strategies are likely a key component of hominins foraging movements since the adoption of a hunter-gatherer lifestyle nearly 2 million years ago in human ancestors[13], playing an important role in human mobility and in our capacity to exploit new environments rapidly and efficiently[21]. Similar approaches to understanding human movements based on the concept of optimal foraging theory have already been invoked for the peopling of Sahul's continental interior[4,22], with major river basins considered the most-attractive environments to human foragers[6]. Yet, the use of Lévy-like movements models has been mostly overlooked when simulating human-population dynamics, dispersal routes, and overall continental sites occupation during initial peopling[23].

In this work, we evaluate the migration across Sahul from a mechanistic movement model[24] assuming a Lévy-like foraging pattern conditioned by time corrected landscape heterogeneity. This latter is determined from a landscape evolution model[25] that characterises the main physiographic changes in Sahul induced by both precipitation and sea-level variations from 75,000 to 35,000 years. Combined with

net primary productivity[7,26], these physiographic changes allow us to rank terrestrial habitats for their impact on human mobility based on evolving landforms, slopes, and the presence or absence of water either in lakes or in primary and secondary drainage basins[6]. To examine the peopling of Sahul, we run two series of 5,000 mechanistic simulations with different entry points (i.e., a northern route through West Papua[10,27] and a southern one from the Timor Sea shelf[2]), calculate the distribution of the produced paths across the landscape, and estimate the migration patterns, likelihood of occupation, and peopling speeds based on location and ages of available archaeological sites[9] (Supplementary Tab. 1). From those simulations, we produce a map of most likely visited regions in Australia. We then demonstrate how our approach could be used as a prospective tool for archaeology and propose potential unrecognised sites across the Australian continent based on predicted dispersal routes and sedimentation patterns.

## Results
### Sahul landscape evolution
To reconstruct the migration of people across Sahul during the Late Pleistocene, we start by characterising the physical environment of the region and its main geomorphic features from 75 and 35 ka. To do so, we use the landscape evolution model goSPL[28] that simulates the joint effects of erosion, sediment transport, and deposition on the relief and drainage network with an adequate resolution (~1 km) to capture geomorphic heterogeneity (Methods). Owing to the relatively modest

impact of geodynamics[29] (~5 m of estimated subsidence in Sahul over the considered period), we assume that the physiography of Sahul is solely controlled by rainfalls and sea level variations. In the model, these variations affect the erosive ability of major drainage systems and enhance channel incisions during low sea-level stands. The simulation is forced with a relative sea-level curve[30,31] as well as a paleo-precipitation dataset, at 2 kyr increments, obtained from a fully coupled atmosphere-ocean-vegetation model (HadCM3[29] – Methods, Fig. 1b, c). As the simulation progresses, the landscape records the concurrent actions of riverine and hillslope processes (Supplementary Fig. 1) and we monitor the changes in erosion, deposition, and rivers hydrology (Methods).

## Sahul environments at the time of human migration

Upon humans' arrival, Sahul geography would have included a mosaic of environments from rainforests, savannas, deserts, alpine regions, grasslands and temperate forests[4,32,33]. Concomitant with their migration, major pressure on existing habitats was related to a gradual aridification induced by the Marine Isotope Stage 3 and characterised by an increase in mean temperature and a decrease in mean annual rainfall (especially after ~45 ka)[34,35]. However, based on calculated precipitation anomalies[36] from each archaeological site (Fig. 1c), hydroclimatic conditions have spatially variable impacts across Sahul. First, we note a decrease in rivers discharge for central Australia after 50 ka associated to a reduction in effective rainfall in both the Lake Eyre drainage basin, the Darling River watershed, and to a lesser extent in the southern Murray River catchment. This decrease is correlated to a reduction in terrestrial sediment flux as the erosive power of simulated streams scales with water flux (Methods – Eq. 1). While continental drying is occurring in central Australia, wetter climates persist in southern Australia, Tasmania, and Papua with stable water and sediment flux over the simulated period. Predicted trends in regional fluvial activity align with patterns of hydroclimatic deterioration (i.e., significant drying trend) and instability (i.e., in palaeomonsoon activity, catchment flow, and lake levels) described from regional palaeoenvironmental proxies (e.g., palynological data[35], sediment cores[32] or palaeoclimate records[34,37]). While those environmental changes would have had local implications on human dispersal and accessibility to permanent water, we find that the averaged mean precipitation per year[36] is sufficient for most of the major drainage systems (even in the most arid central regions) to be intermittently active and connected periodically during inundation events (Supplementary Fig. 1). Interpretations based on modern distribution of permanent water points across Australia[6] suggest that similar distribution and connectedness of water conditions likely existed during the Pleistocene. We also note that reconstructed stream paths and associated drainage basins remained stable with no major drainage reorganisation and with river captures restricted to low topographic gradient regions (i.e., along coastlines and around Lake Carpentaria). We attribute this stability to the relative quiescence of the regional tectonics[29], exemplified by the similarities between predicted drainage divides and present-day hydrological basins (presented in Supplementary Fig. 2).

## Mechanistic movements conditioned by landscape heterogeneity

To account for physiographic impacts on human dispersal across Sahul, we combine four physiographic layers derived from our landscape simulation outputs (slope, environmental and physiographic diversity indices, position of stream flow, and water fluxes – see Methods, Supplementary Table. 2, and Supplementary Fig. 2) known to influence early human migration[6,9,10,38–40]. In addition, a regional net primary productivity layer is defined from LOVECLIM climate reconstruction[7,26] as an indicator of relative ecological carrying capacity of the ecosystem[7,41] (Supplementary Fig. 2) and low cost areas are

enforced along coastal regions[42,43]. These layers are then amalgamated to create a normalised physiographic resistance map that defines how amenable a region is for mobility, with high values representing regions hindering movement (Supplementary Fig. 3). This resistance map then defines the environmental conditions for the mechanistic model of ecological displacement SiMRiv[24], in which the migration trajectories of humans are conditioned by landscape complexity, perceptual range (set to 10 km with a maximum step length of 1 km, see Methods), and partial memory of past displacement (semi-correlated Lévy-like foraging patterns[44,45]). Applying this approach to the long term dispersal of early humans across Sahul, from a northern route located in the Bird's Head Peninsula in West Papua[8,10,27] and a southern one via the Timor Sea shelf into northwest Australia (Methods – Supplementary Fig. 4), we opt for a probabilistic assessment of a large set of simulations (i.e., for each route we run 5000 realisations over 10 million steps, where the latter is chosen based on estimated human generations required for the peopling of the region[8] and mechanistic movement step length – Methods)[23]. We then combine the different predicted paths to evaluate the travel patterns and estimate the likelihood of humans moving via any given location within the region. This combination of predicted migration paths and likelihood of movement is used below to infer the distribution of human occupation across Sahul prior to 35 ka and their associated migration speed based on archaeological datasets[9] (Supplementary Table 1 and S3).

## Migration patterns and residential likelihood

Hunter-gatherer populations are highly subject to ecological and environmental constraints that structured their use of space and their mobility[8]. By incorporating landscape influences and ecological carrying capacity on local behaviour, our mechanistic movement simulations aim to mimic movements of groups of individual[19]. As previously mentioned, the approach accounts for different types of displacement not solely related to migration, such as those associated with subsistence[21,46,47]. This reflects that the majority of the time individuals spends moving is not spent in moving camp, but in logistical forays to hunt or to gather plant food. Commonly reported ethnographic patterns[4,48,49] show ranges from 2 to 15 km in daily foraging movements from base camps. However, distances might vary significantly based on environmental conditions[50] and daily hunting distances of up to 25 km or more are possible[51]. The relationship between environmental conditions and daily hunting distance is not straightforward, for example it was not uncommon for some Australian Aboriginal people to accept extremely low caloric intakes and forage up to 15 km from camp rather than move from a secure water source[52].

Extracting residential moves from foraging ones in simulated mechanistic movements requires caution. In part, this is due to the difficulty in obtaining such data from ethnographic sources but also because when such data exist, they are often normative, inferential, or extrapolated from one season to another[52]. In addition, compiled data from Australian Aboriginal people exhibit large differences in the number of residential moves per year and in the average distance between each residential camp; likely highlighting the role that ecosystem net primary productivity played on hunter-gatherer mobility[53]. The Anbarra people from Arnhem Land, for example, move only a few times a year and over relatively short distances (~3 km)[52], while some groups of Ngadadjara foragers in the Australian western desert could change camp more than 30 times per year with average distances of up to 40 km between each residential base[52,54]. To estimate the number of residential moves out of the Lévy-walk simulations, we aggregate successive displacements (combining random and correlated steps) made within a specified radius and consider them as part of a single mobility event in their migration. The selected radius implicitly represents the knowledge a group of individuals has of its surroundings but also its willingness to move across the landscape informed by

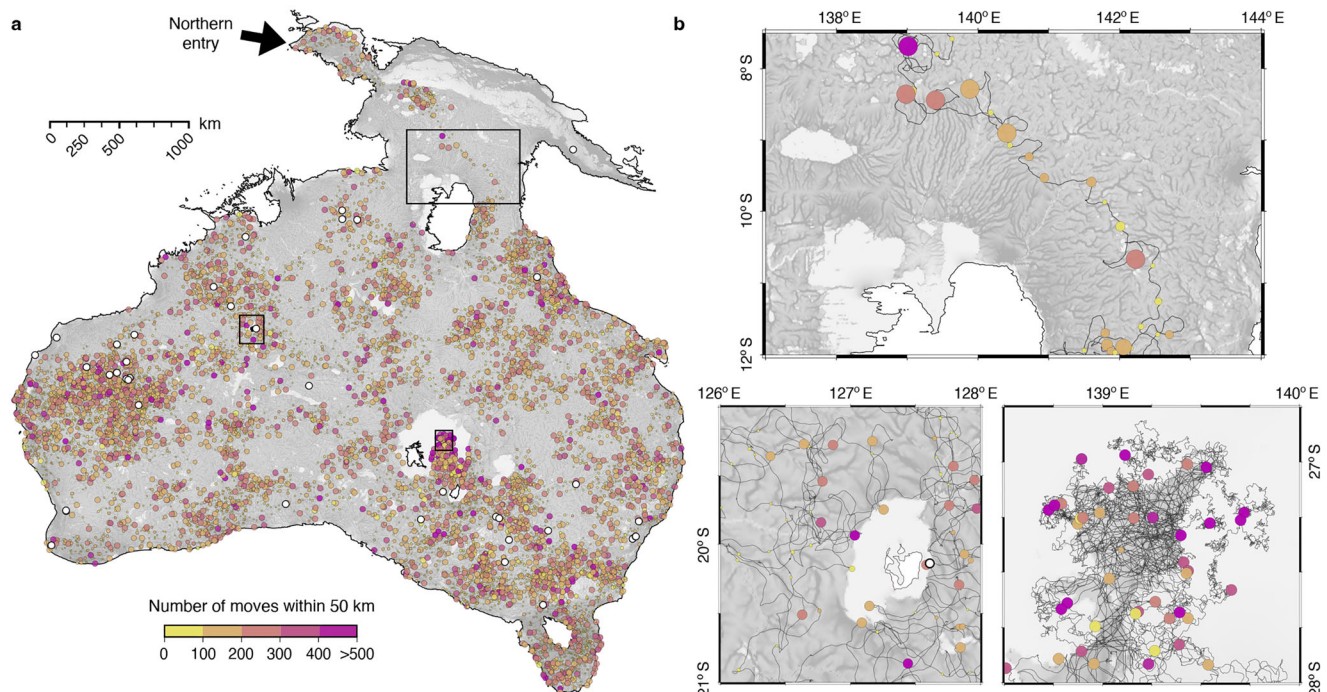

**Fig. 2 | Characterising migratory patterns from two-state Lévy-like walkers.
a** Results from one of the mechanistic simulations showing predicted migration routes passing through 34 of the 40 archaeological sites defined in Supplementary Table 1. Colours represent the number of walker moves between consecutive circles and the size of the circle is scaled based on the cumulative distance travelled by walkers for each 50 km segment. **b** Zoom in three regions illustrating different patterns of movement from relatively direct migration in the area surrounding Lake Carpentaria (governed by the correlated random walk state) to more erratic behaviour (lower right panel) forced by the underlying high-cost surface of the Tirari (left) and Strzelecki (right) deserts. Generated paths using the SimRiv software and visualised with the open-source python interface for the Generic Mapping Tools (https://www.pygmt.org).

environmental and carrying capacity constraints[7]. For the reasons explained above, it seems unlikely that the choice of a unique radius can be representative of the overall migration across Sahul. Instead, we test two distances set at 25 and 50 km respectively. Figure 2a illustrates the corresponding aggregated walkers' pathway for a single mechanistic movement realisation starting from the northern entry with a radius of 50 km. This approach also indirectly tracks the human activity/energy from one migration point to the other by recording the cumulative distance travelled and the number of random and correlated walks between residential moves. Straightforward trajectories with long steps thus correspond to groups with high residential mobility across regions of low physiographic cost (Fig. 2b - top panel) while small travelled distances between consecutive migration steps containing a large number of small movements either apply to groups of hunter-gatherers with limited residential mobility but high logistical mobility[52,55] or correspond to regions with high physiographic cost impeding movement (Fig. 2b – bottom panels).

For each realisation (that tracks the position of a single group of individuals starting from one of the two entry points), we evaluate (1) the likelihood of occupation of all parts of Sahul during the transient phase of dispersal and (2) if the archaeological sites defined in Supplementary Table 1 are within any walker reach (we consider that a site is accessed if inside a 10 km radius from a walker). The best performing mechanistic realisations have up to 39 out of the 40 archaeological sites visited for each considered entry point (Supplementary Fig. 4). Because simulated movements might repeatedly pass in the vicinity of a site, we first extract all the walkers close to a site for any particular simulation (Supplementary Fig. 5) and then group them based on the calculated cumulative travelled distance to the initial entry point. Grouping is done using a clustering *k-means* method[56] and provides an estimate of the number of distinct visits for every site. The analysis is systematically performed for each mechanistic movement simulation

(5000 realisations for each entry point) and the combined results are presented in Fig. 3.

Likelihood of occupation for individual realisation shows a wide range of distribution with regions of high kernel density estimates (KDE) found over the entire continent (Supplementary Fig. 5a). When combined, the predicted high occupation areas vary significantly between the chosen *arrival into Sahul* locations. Unsurprisingly, for the scenario with an entry point on Bird's Head Peninsula[8,10,27], northern regions exhibit the higher KDE (Fig. 3a). The occupation distribution then shows two preferential dispersal routes on both sides of Lake Carpentaria (more significant on the eastern side of the lake – possibly related to higher net primary productivity as shown in Supplementary Fig. 2) that then propagates towards the continental interiors. When considering a dominant southern route through Timor into Kimberley region[2,8], the occupation distribution reveals much higher likelihood into the western interiors spreading South of Lake Carpentaria towards the East as well as within the southern regions of Western Australia (Fig. 3b). While few of the most peripheral coastal sites in Western Australia show limited occupation when considering the northern route scenario (Boodie Cave, Devils Lair and Jansz – sites 3, 8 and 13 in the Supplementary Table 1), the other ones have normalised KDE values above 0.20 with more than half (21 sites) reporting values above 0.35. Those values are much higher when considering the southern route hypothesis (KDE > 0.4 for 38 out of the 40 sites – Fig. 3b). The percentage of walkers arriving on each site (Fig. 4a) shows that all of them are reached with values above 30% (i.e., more than 1500 realisations predicted at least one visit to these sites), except for the Ivane Valley in the New Guinea highlands[57] (<5% for both entry points, site 12 on Figs. 1a and 4a). For the northern entry, the following two sites with percentages ≤45% are Boodie Cave on Barrow Island and Jansz Cave on Cape Range Peninsula (both located along the Western Australia coastline – site 3 and 13 respectively on Fig. 4a). For both entry points,

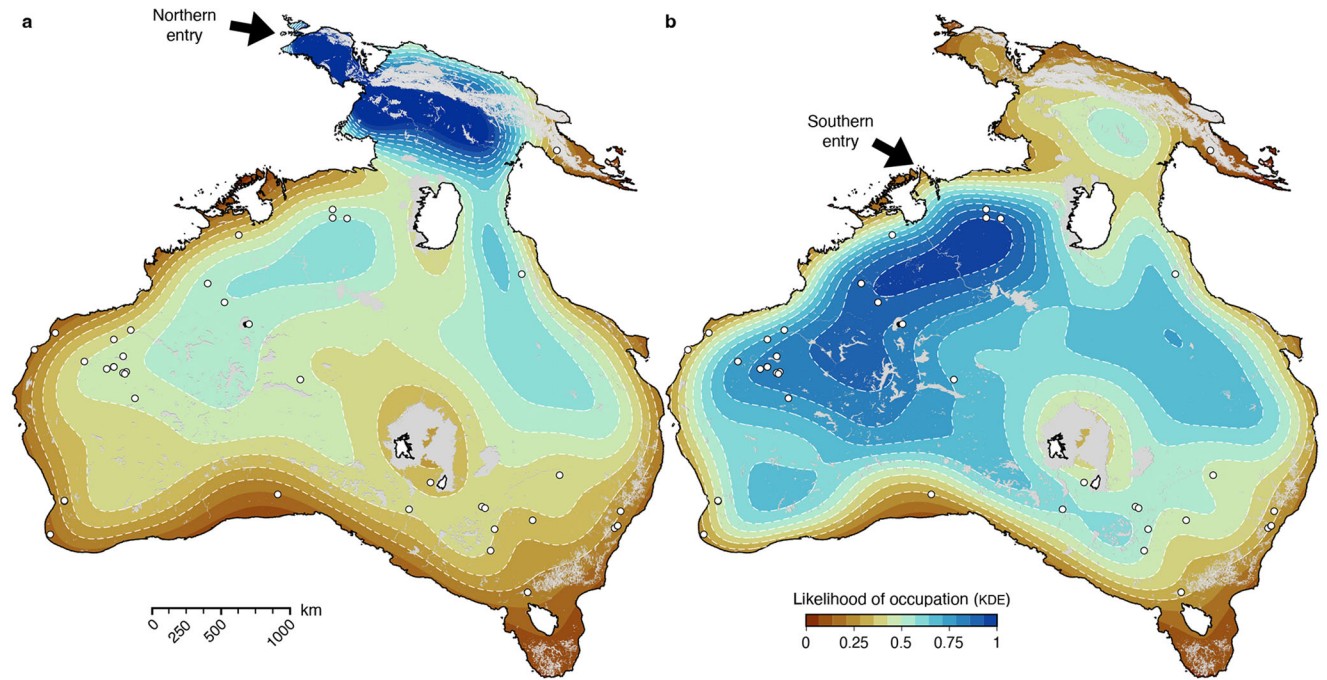

**Fig. 3 | Analysing the distribution of human occupation across Sahul from mechanistic simulations.** Statistical analysis of the occupation of the region by early humans where the kernel density estimation (KDE, normalised) indicates the likelihood that humans came across a given area. **a** KDE results considering an entry point into Sahul from Borneo to Misool in northwest Papua then into the Bird's Head[10,27,31]. **b** KDE results for a southern route through Bali and Timor into the modern-day Kimberley region of Western Australia[2]. White circles represent archaeological sites[9] (Supplementary Table 1). Maps are produced with the open-source python interface for the Generic Mapping Tools (https://www.pygmt.org) based on paths generated with SimRiv software.

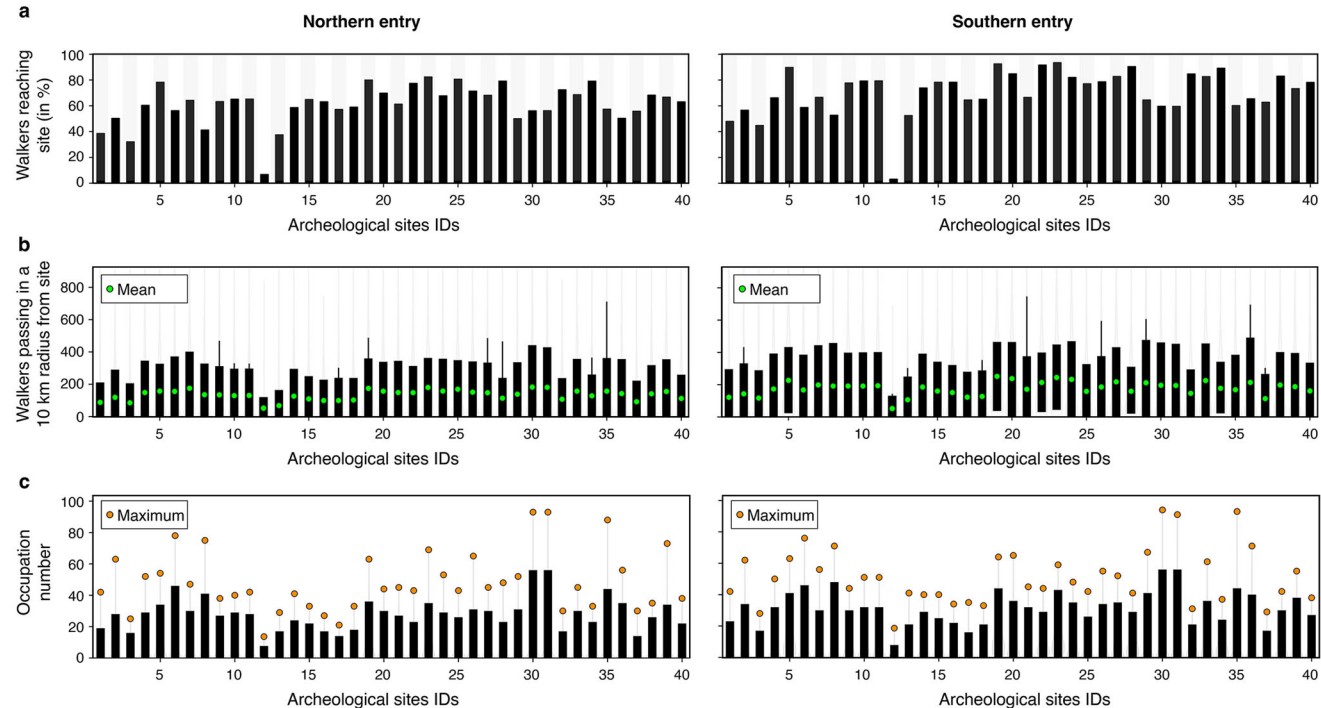

**Fig. 4 | Predicted walkers reaching archaeological sites and associated sites occupations from mechanistic simulations.** **a** Distribution of walkers around chosen archaeological sites[9] (Supplementary Table 1) for the northern and southern entry points (left and right respectively) combining 5000 simulations in each case and showing percentages of time the sites are visited. **b** Analysis of the number of walkers in a 10 km radius around each site extracted by counting the walkers per simulation and showing the distribution extent and its mean (green circle – calculation is performed for each route). **c** Associated occupation number representing distinct visits to sites for each route. The distribution is calculated for the 5000 simulations after extracting the number of visits for each realisation (orange circle presenting the maximum number of visits).

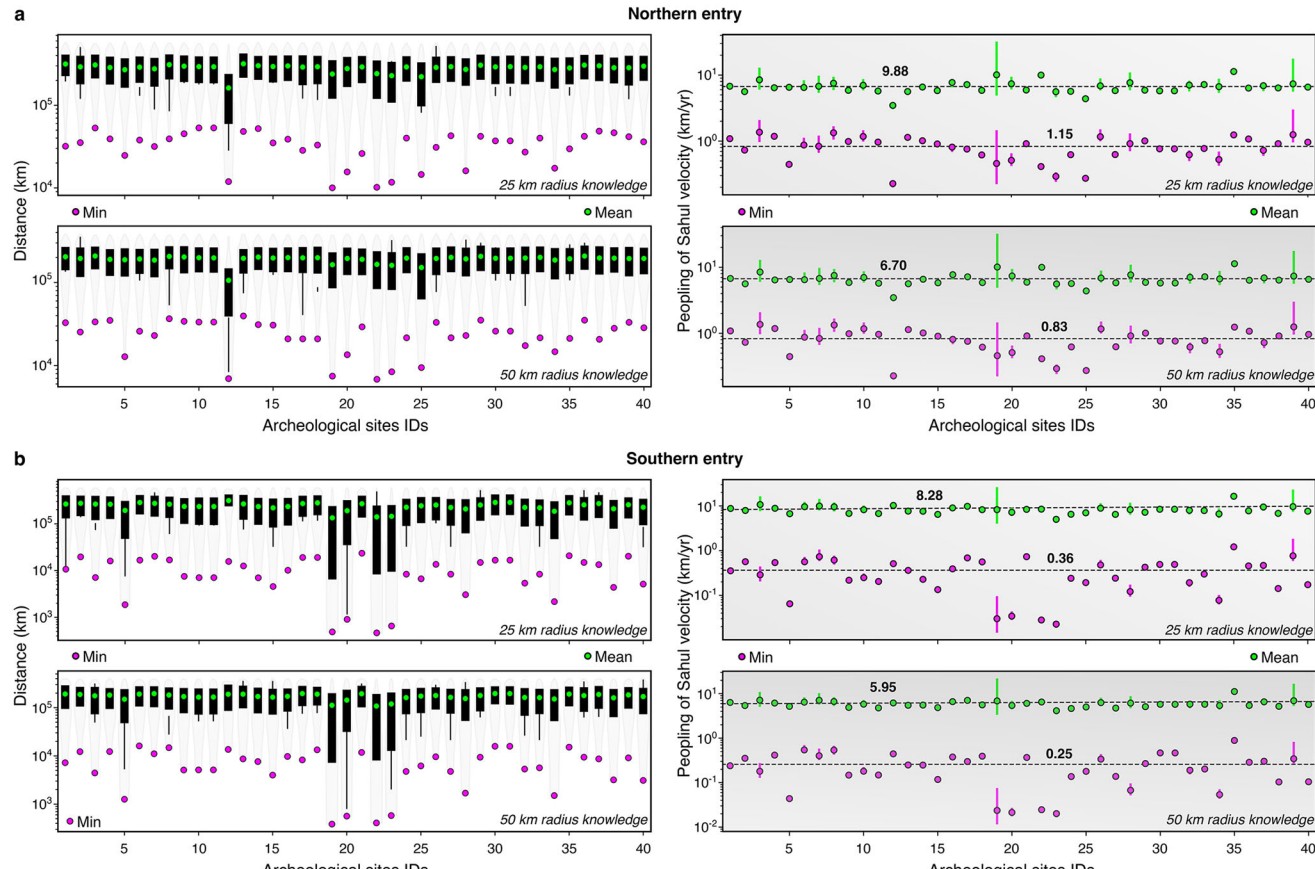

**Fig. 5 | Travelled distances and migration speed across Sahul.** Predictions of travelled distances and speed from the Bird's Head region[10,27,31] in New Guinea to archaeological sites are presented in (**a**). Similar calculations are performed for the southern entry point into the Kimberley region (Western Australia)[2,10] and shown in (**b**). For each entry point, two scenarios of early humans' knowledge of their surrounding environment are estimated (i.e., a 25 km radius of landscape awareness and a 50 km radius). The corresponding distances travelled to reach each site are statistically derived and presented as box plots on the left panels. Estimates of migration speed, on the right panels, assume a first arrival time at 73 ka[8] for the northern route and at 75 ka for the southern route[2]. Humans peopling velocities are based on the predicted travelled distances to archaeological sites and the dating age ranges provided in Supplementary Table 3. For the 40 archaeological sites, the sampling size n for the northern and southern routes is obtained from the 5000 paths simulated with each path composed of 10 million walker steps aggregated assuming a 25 or 50 km radius of knowledge. The bounds of the box show the 25 and 75 percentiles in the right panels with the mean and minimum values defined by the green and fuchsia circles respectively. The velocities on the right panels are obtained from the mean and minimum obtained in the left panels assuming a first arrival time and the age ranges provided in Supplementary Table 3.

three of the iconic sites on the edges or top of the Arnhem Land plateau[1,33], namely Madjedbebe, Nauwalabila and Nawarla Gabarnmang (sites 19, 22 and 23 – Fig. 1a, b) have percentages above 60% combined with average number of walkers >200 and occupations values > 20 (Fig. 4c). The other sites with highest percentage of walkers (>70%) are found at Ngarrabullgan Cave (site 25) in North Queensland, the Puritjarra rock shelter (site 32 in Central Australia), the Riwi Cave (site 34) and Carpenter's Gap 1 rock shelter (site 5) in the Kimberley (Western Australia). Predicted number of walkers (i.e., the average number of times walkers are in the vicinity of a site for each realisation) range between 100 and 500 with inferred occupation numbers varying between 10 and 30 for most sites (Fig. 4c), with the highest values found on the banks of the Nepean River in New South Wales (occupation number >50 – sites 30 and 31)[58].

### Travelled distances and peopling speed of Late Pleistocene Sahul

Along with the migration patterns and occupation analysis, we also extract the cumulative distances travelled by walkers to reach each archaeological site (left panels on Fig. 5a, b). As already mentioned, our approach accounts for both residential mobility and logistical mobility that both vary along several potentially independent behavioural and

environmental dimensions (e.g., the forager–collector model[59]). To evaluate migration speeds across Sahul, we extract residential mobility values by specifying two radius lengths for hunter-gatherer average residential mobility distance, one set to 25 km and the second equals to 50 km. Each value corresponds to the extent of a long-distance dispersal journey above which we consider that a walker has entered a previously unexplored region or a different environment that would require the establishment of a new base camp. Estimates of long-distance dispersal lengths have been proposed with values ranging between 22.4 and 69.3 km based on omnivorous and herbivorous mammals allometric relationships[8,60]. The lower length (25 km) is within the reported residential mobility values of average African hunter-gatherers and Australian Aboriginal people (21.1 to 35.2 km)[8,52], while the higher one (50 km) reflects the fast mobility of the reported migrations of up to 60 km in six months[54]; exemplified by the high mobility of Ngadadjara people[52]. Using these two radii, we extract travelled distances to each site and each entry point (left panels in Fig. 5). For the northern route, we find minimum values between -10⁴ and $6 \times 10^4$ km for a 25 km radius and varying between $6 \times 10^3$ and $4 \times 10^4$ km for the larger radius (50 km). Predicted minimal travelled distances for the southern route show lower ranges from $4 \times 10^2$ to $3 \times 10^4$ km and from $3 \times 10^2$ to $2 \times 10^4$ km when considering the 25 and

50 km radius respectively. As expected, an increased residential mobility (higher radius length) with foraging and hunting activities (i.e., logistical mobility) taking place along the way facilitates the migration across the region and would be congruous with shorter travelled distances to each archaeological site (Fig. 5). Overall, mean travelled distances comparisons show marginally lower values for the southern route ($2.32 \times 10^5$ and $1.91 \times 10^5$ km when considering the 25 and 50 km radius respectively) than for the northern route ($2.47 \times 10^5$ and $1.94 \times 10^5$ km).

From the predicted walked distances and available dating of the first occupation of each archaeological site (minimum, mean and maximum dated ages—Supplementary Table 3), we estimate the migration speed of Sahul assuming a first arrival time as early as 75 ka for the southern entry and followed by the northern entry at 73 ka[2,8] (right panels in Fig. 5). When considering the mean travelled distances, we found average mean values across all sites of 8.28 and 5.95 km/yr for the southern entry and of 9.88 and 6.70 km/yr for the northern one, depending on the assumed knowledge radius (i.e., 25 or 50 km respectively). These mean velocities drop to 0.36 and 0.25 km/yr for the southern entry and to 1.15 and 0.83 km/yr for the northern one if the average minimum travelled distances per site are used (Fig. 5). Regardless of the chosen radius, calculated migration speeds represent the lowest velocity range estimated from travelled distances for two reasons. First, the dates are minimum peopling ages: they do not necessarily represent the earliest presence of people at a site and the sites may not represent the first presence of people in a region. Second, peopling of Sahul was likely accomplished in successive waves, unrestricted to the chosen northern and southern entries[2,8,10,27] nor to a 75–73 ka double-entry scenario[2]. Despite this, the predicted peopling speeds are within the range estimated for other regions and with other approaches. As an example, northern Europe hunter-gatherers' migration speed ranged between 0.7 and 1.4 km/yr[61] while estimation from North America Clovis-age occupations suggests values of 5 to 8 km/yr[46]. Similar rates of 6 to 10 km/yr have been proposed from archaeological evidences[62] for the first peopling of the Americas, and the Neolithic transition in Europe. Analysis of mtDNA variations across south-east Asia populations suggests a dispersal of Homo sapiens from Eurasia to Australasia at speed of 4 km/yr[63]. Stochastic-ecological models support a peopling of the entire Sahul continent at a rate of 0.71 to 0.92 km/yr[8]. Finally, recent work[2] based on directionally supervised cellular automaton model suggest movement rates of the order of 0.4 km/yr.

## Discussion

### Predicted peopling of Sahul

Our findings provide an estimate of the direction and speed of human dispersal in Sahul, based on hunter-gatherer movements forced by physiographic complexity. As such we do not explicitly capture the role that demographics of the travellers might have played, nor do we account for specific rules related to social, cultural, and economic decisions[1,6,64,65]. Despite the limitations, analysis of migration and occupation patterns from our results shows that all known archaeological sites are visited by modelled walkers with calculated peopling of Sahul velocities within the range proposed in other studies[2,8,63]. No statistically optimal superhighways[2,9] nor major preferential coastal routes[42,43,66] arise from our KDE. Instead, it either advocates for a spread on both sides of Lake Carpentaria[4,6] along major drainage systems when considering a northern entry (Fig. 3a) or a radiation towards South and East following perennial rivers and streams on the western side of Australia for the southern entry (Fig. 3b).

At the time of the first peopling, palaeoecological reconstructions show evidence of a strong diversity of interior habitats varying from high-altitude tropical forest-grassland, subtropical savannah, to semi-arid woodland and grassland[67]. The idea that humans dispersed rapidly through the interior foraging along water streams has been

hypothesised in the past[4,6,22]. As an example, reported Warratyi rock shelter (site 37 – Supplementary Table 1) archaeological evidences demonstrate the presence of humans in the southern interior of the continent by 46 ka[65]. Our results support the hypotheses of Hiscock and Wallis[68] and Veth et al. [69] that Aboriginal people would have already settled in the Australian arid interior by the time more extreme conditions over the last glacial maximum (35 to 15 ka)[34,37,70] transformed many of these interior habitats with the expansion of potential environmental barriers (e.g., sandy deserts)[65,71,72] and variable biological productivity[70].

To compare our mechanistic movement simulations with recent works based on least-cost pedestrian travel modelling[9] and supervised cellular automaton[2], we compute the probability of presence in Sahul (counting the number of walkers found in 0.05° cell size – Fig. 6) assuming a double-entry scenario (northern and southern routes – see Supplementary Fig. 6 for the predicted heat map of Australia when considering each entry point separately)[2]. We find that probability of walkers' presence is high in the vicinity of the major corridors[9] (Fig. 6). This is specially the case for the migration corridor east of Lake Carpentaria that follows the Great Dividing Range, the southern corridors connecting Lake Eyre to the eastern corridors, the central superhighways transecting Australian arid interior, as well as several primary and secondary pathways found for the Kimberley, Pilbara, and Arnhem Land regions[9]. Yet, we also identify some inconsistencies between the superhighway model[9] and our predicted probability of presence. As an example, our mechanistic simulations do not favour a northern corridor passing through the New Guinea Central Cordillera and Eastern Highlands up to the Owen Stanley Range (Fig. 6). Instead, our probability map suggests higher migration probabilities across New Guinea foreland basins with limited ability for migration to high elevations (e.g., lowest predicted walkers and occupation number found in the Ivane Valley[57] – site 12, Fig. 4a). While our predictions suggest moderate to high probabilities along the south-western coastlines, the proposed superhighway across the Australian Bight[9] is not expected in our approach. Here, we relate the limited probability of walkers' presence to the small number of paleo-rivers, low physiographic diversity index and restricted net primary productivity of this region (Supplementary Fig. 2a). In addition, those differences might be induced by the choice of a double-entry scenario[2] where the superhighways model adopts a more holistic approach that uses different sets as origins or destinations[9] (e.g., grid-to-grid, grid-to-coastline and grid-to-water). The fact that there is good alignment in many areas, despite only using two entry points, is very encouraging and suggests that our approach can reproduce some of the major migration routes proposed in the superhighways model[2].

### Foraging and migrating across transient landscapes

Compared to other approaches, there are two significant improvements in the methodology that we propose. First, while some approaches consider the underlying complexity of the landscape when evaluating the spread of humans across Sahul[2,9], they mostly assume static landscapes, and do not consider the impact of climate-driven geomorphic changes taking place during the time of migration[6]. Yet, at both catchment and regional scales and on relatively short time frames, surface processes can drive significant modifications in drainage network flux and organisation which would influence both the speed and direction of dispersal[23,73]. By accounting for spatially varying geophysical attributes influenced by climatic evolutions (Fig. 1), this study relies on more realistic topographic and environmental constraints (Supplementary Fig. 1) to model human interactions with the terrains and environments they live on (Fig. 2 and Supplementary Fig. 4). Changes in these parameters could either facilitate or limit travellers' expansion into new regions by modulating the pattern of physiographic resistance to displacement (Supplementary Figs. 2 and S3), ultimately impacting on the speed of dispersal. Models

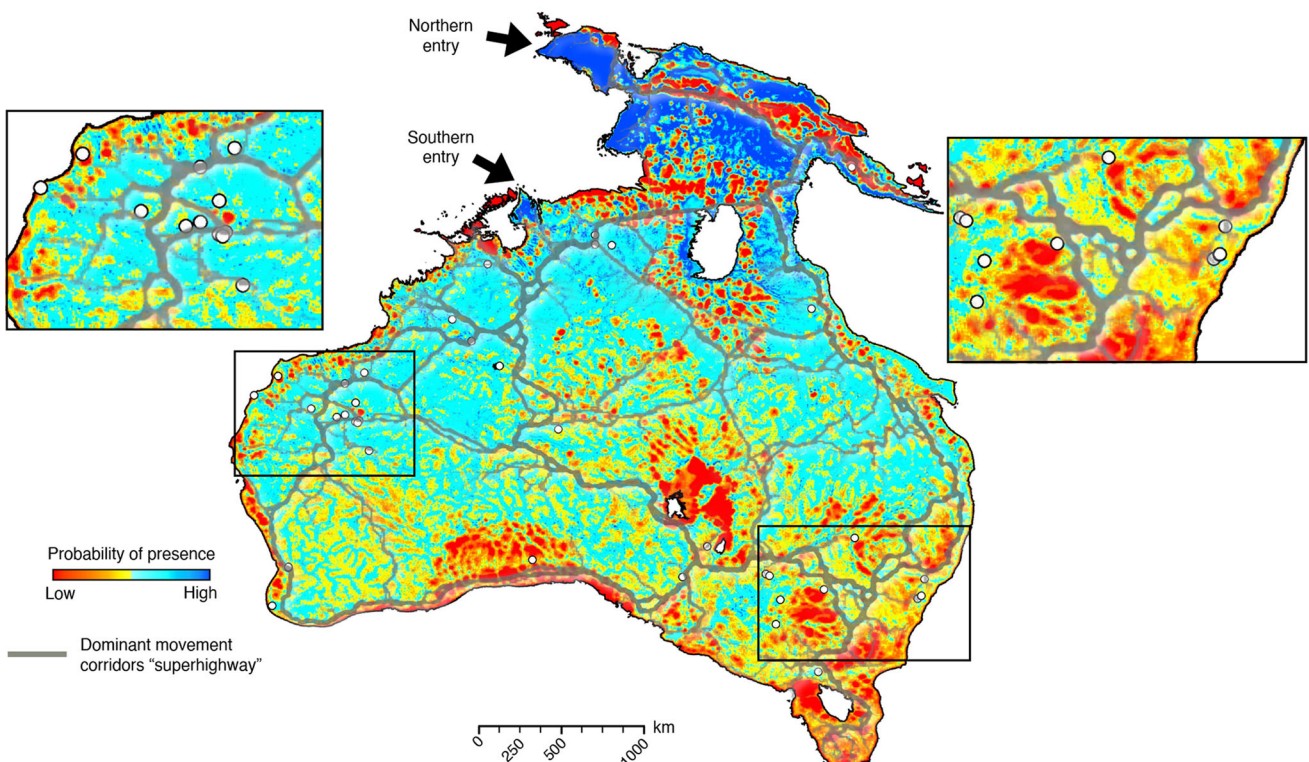

**Fig. 6 | Probability of Lévy walkers' presence across Sahul combining the northern and southern entry points.** Heat map of Sahul walkers' distribution estimated by merging the results from the mechanistic simulations (two entry points each based on 5000 realisations with 10 million walker steps by realisation). The number of walkers from all simulations are aggregated in 0.05° cells (~5 km). Cyan to blue areas correspond to regions with high probability of presence and red to yellow areas are regions which are never or rarely chosen. White circles indicate locations of archaeological sites (Supplementary Table 1). Grey lines overlaying the map show the dominant movement corridors calculated with least-cost pedestrian travel model[9] and interpreted as the major pathways - superhighways - of human migration across Sahul prior to 50 ka[2,9]. Maps are produced with the open-source python interface for the Generic Mapping Tools (https://www.pygmt.org) based on paths generated with SimRiv software.

not explicitly accounting for physiographic changes are likely to misjudge the travelled distances during migration as well as the timing of early human expansion. Our statistical analysis of simulated travelled distances provides a range of plausible mean spreading rates varying from ~6 km/yr up to ~10 km/yr. As previously mentioned, the approach does not assume any travel time and velocities are obtained *a posteriori* based on walked distances from chosen entry points and archaeological dates (Supplementary Table 3). We find that the obtained rates are within the speeds reported in other studies[8,61,63] with lower mean spreading rates (from 0.25 up to 1.15 km/yr) close to the ones derived from stochastic-ecological models[8] (0.71 to 0.92 km/yr) and supervised cellular automaton models[2] (0.4 km/yr).

The second improvement lies in the incorporation of Lévy walk foraging hypothesis in our simulation of human dispersal across Sahul. Lévy walk has been identified in the movement patterns of human hunter-gatherers[19,20] and has been mathematically proven to be an optimal method when searching for food, outperforming foragers using other kinds of movement patterns[18,74]. It not only describes short and medium length movements within specific patches, but also reproduces rapid, long-distance migrations[21] based on foraging and mobility characteristics[52] as well as differences between residential moves across Aboriginal groups[52,54]. As a result, the proposed Lévy walk foraging framework complements well existing pedestrian-transportation network models which account for ethnographic and demographic data[2,7,9] and that have been previously used to evaluate used to evaluate the timing and spread of human dispersal in Sahul[2,9]. If one assumes that early humans foraging indeed followed Lévy walk movements, estimates of mean travel times between environments (a

cornerstone in standard formulation of migration models) could not be directly inferred[75] and more realistic assumptions should be implemented to better characterise human mobility across heterogeneous landscapes[21]. In addition, Lévy walk movements do follow a power function which suggests that the typical assumption of a Gaussian distribution of migration distances often used in migration-diffusion approaches may need to be reconsidered[13,15,16].

### Landscape modelling as a prospective tool for archaeology

An interesting outcome of the approach lies in our ability to evaluate residential likelihood across the region. To demonstrate it, we randomly position over the Australian continent 5000 hypothetical sites and perform a similar analysis to the one described for the known archaeological sites[9] (as in Fig. 4). Using the previously run 5000 mechanistic movement realisations starting from the northern entry point[10,27], we first calculate travelled distances for each of these hypothetical sites and produce a map of mean walked distances for Australia (bottom left panel in Fig. 7). This predicted map suggests that human settlers would have dispersed predominantly across the continental interior along rivers and river basins corridors on both sides of Lake Carpentaria (Fig. 3) and highlights a preferential north-south route rather than a coastal one[42,43,66]. Yet, we find some regions with relatively low walked distances along the northern and central coastlines of Western Australia, the Great Australian Bight (South Australia), and in Queensland (Fig. 7). Limited occurrences of coastal routes on the most southern parts of Western Australia and in southern Victoria are partly related to the low regional physiographic diversity of these now submerged areas on the southeastern edge of Sahul continental shelf (Methods – Supplementary Fig. 3).

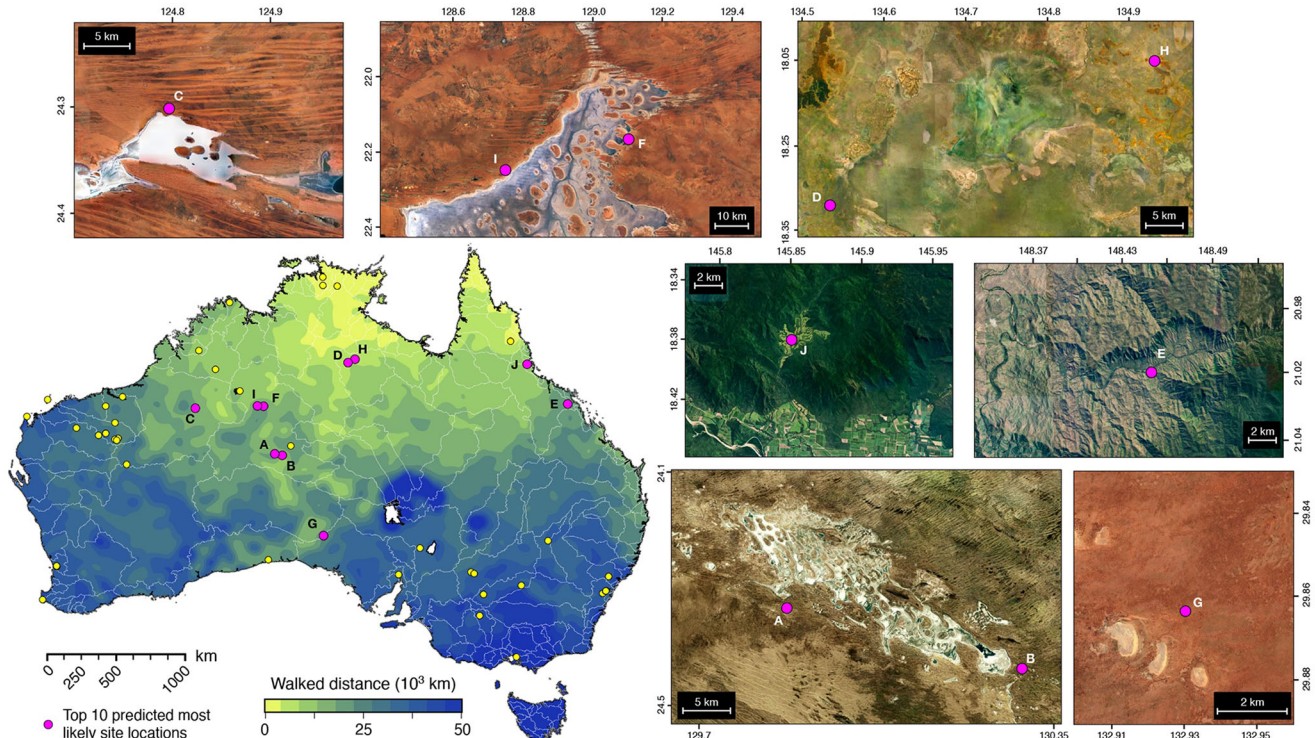

**Fig. 7 | Identified areas with archaeological potentials.** Predicted minimal walked distances assuming a northern entry point and computed using 5000 sites randomly distributed across Australia landmass (bottom left map – white contours show present-day hydrological basins). From the percentages of time each of those random sites are visited as well as the number of walkers in their vicinity (similar analysis to the one performed in Fig. 4); the 10 most likely residential sites are extracted (purple circles from A to J – Supplementary Table 4). Satellite images show the locations and environments of those predicted archaeological sites (imagery obtained from Digital Earth Australia https://de-australia.terria.io/ and are available under Creative Commons Attribution 4.0 International Licence http://creativecommons.org/licenses/).

In a second step, we extract the 10 most likely sites to be occupied (ordered from sites A to J in Fig. 7 and Supplementary Tab. 4), on the basis of having the greatest percentages (i.e., visited more than 70% of the time from the mechanistic realisations) and the highest number of walkers in their surroundings (>200). Those Australian locations illustrate some areas that have not been the focus of substantial archaeological attention and provide a perspective on the Pleistocene archaeology of Sahul. The identified areas are distributed across a wide range of environments (Fig. 7) from wetlands in the Northern Territory (sites D & H), to the banks of perennial and ephemeral rivers and lakes across the Pilbara or Western Desert regions (sites A, B, C, F & I), to places in the proximity of waterholes in arid region (site G), and to incised valleys near the coast in the Atherton Tablelands region (sites E & J). The archaeological database (Supplementary Table 3) equally shows significant environmental differences between locations with sites in the arid interior close to relict overflow systems (e.g., Willandra Lakes region[76]), in sandy, shield and stony deserts characterised by internal and uncoordinated drainages and water holes (e.g., Western Desert East of the Pilbara uplands[71]), or along confined sandstone gorges with abundant fluvial activity in lowland basins (e.g., Cranebrook Terrace[77]).

Our method does not consider archaeological site taphonomy but the results from the landscape evolution model could be used to evaluate the morphological changes around each site (Supplementary Fig. 7), and thereby serve to infer the preservation potential at given locations that could then be further investigated with refined stratigraphic frameworks (i.e., combining sedimentological and soil micromorphological analysis[1,65,76,78]). It is noteworthy that all sites (with the exception of site G) are located where predicted landscapes experience substantial changes (Supplementary Fig. 7). Some of the predicted sites located close to ephemeral rivers are subject to repetitive flooding episodes and exhibit relatively high deposition rates that may have facilitated the preservation of artefacts. Others, in the Atherton Tablelands region, are characterised by significant erosion rates that could impede the preservation of possible artefacts. We hypothesise that sites I and F on the edges of the Lake Mackay (also named Wilkinkarra by Pintupi people), an ephemeral lake in the Pilbara, are worth investigating due to its limited regional erosion and deposition. Occupation in similar environment around 45 ka has already been reported in the Parnkupirti site[79] (site 28 – Supplementary Table 1) at Lake Gregory ~200 km north of the proposed sites. For similar reasons, site G south of Tallaringa Conservation Park may have unexplored archaeological potential. Here again, the role that small permanent water bodies such as waterholes could have played in the dispersal of original inhabitants across arid and semiarid regions has been proposed in the past[6]. Incorporating a refined temporal description of interrupted (e.g., the Western Desert[71]) versus continuous drainage systems (e.g., Lake Eyre Basin and Willandra Lakes[76]), by integrating site specific variations inferred from local proxies[32,33,35], would further improve our predictions. Nevertheless, by integrating explicitly residential and logistical mobility patterns with physiographic changes, the approach presented here offers testable hypotheses that could be used predictively to identify new archaeological survey campaigns and improve our understanding of the dispersal of modern humans. Because the resolution of our simulations is at 2 km, we envision that our approach could provide valuable and cost-effective insights at regional scale on sites residential likelihood prior to archaeological fieldwork.

In this study, we propose an approach to evaluate the spread of human dispersal in Sahul by combining evolving physiography attributes and a mechanistic model of hunter-gatherer movement that encapsulates stochastic and decision-making behaviours. Our method

focuses on an optimal technique for searching for heterogeneously distributed food that mimics the foraging patterns observed in contemporary human hunter-gatherer populations[19,21]. Populations whose actions and decision-making mechanisms were ingrained with cultural knowledge and traditions. Our simulations highlight peopling in all directions from the arrival point, with preferential movement throughout the continental interior (north-south and west-east routes) following resource availability along both perennial rivers (the Carpentaria, Lake Eyre, and Murray-Darling Basins) and ephemeral streams. We evaluate the ensemble of realisations by analysing the predicted distribution of human occupation against archaeological sites[9] and by inferring the peopling velocity (migration speed) across Sahul. Calculated migration speeds are within the range obtained for Sahul using complementary approaches[2,8]. It also shows that all sites are visited (with predicted occupation percentage generally above >40%) suggesting repeated (likely short-term) camp sites by highly mobile populations rather than large residential settlements. This could explain the general pattern found in older archaeological sites across Sahul which are mostly small and relatively shallow with a limited number of artefacts[4]. We believe our approach complements existing models[2,4,8,9] and offers an original view that can be used predictively to identify potential unrecognised archaeological sites in Australia and elsewhere in the world.

## Methods

### Physiography modelling

In this study, we use goSPL[25,80] an open-source scalable parallel numerical model that simulates landscapes and sedimentary basins evolution across a variety of spatiotemporal scales[81]. goSPL relies on an implicit and parallel algorithm to solve its constitutive equations[82]. It accounts for river incision and soil creep, both considered as the main drivers of long-term physiographic changes. Series of forcing could be used to inform on region-specific geo-history such as spatially and temporally varying tectonics (both horizontal and vertical displacements) or multiple climatic conditions (e.g., precipitation patterns and sea-level fluctuations).

A multiple flow direction algorithm routes water and sediments for upstream catchments to their depocenters. Unlike single direction approach, it prevents the locking of rivers into unique pathways. As a result, it allows for a better representation of (1) low-relief natural river system (such as branching river networks) and (2) the distribution of sediment flux in downstream regions. For the continental domain, the model's main equation, the continuity of mass, has the following common form:

$$\frac{\partial z}{\partial t} = U + \kappa \nabla^2 z + \epsilon P^d (PA)^m \nabla z^n \tag{1}$$

Equation 1 links changes in surface elevation ($z$ in m) with time ($t$ in yr) to (1) tectonic forcing ($U$ in m/yr and here set to 0), (2) hillslope processes defined using a standard diffusion equation with $\kappa$ the diffusion coefficient (in m²/yr)[83], and (3) fluvial processes expressed using the stream power law (SPL). The SPL is an empirical law that quantifies the rate of incision based on local slopes and the amount of water passing through a cell. The contribution of the slope ($\nabla z$) and water flux ($PA$) to erosion is scaled based on two dimensionless exponents ($m$ and $n$ – set to 0.5 and 1 in this study). The coefficient $\varepsilon$ is a precipitation-independent erodibility value (in /yr based on the choice of $m$) that can vary spatially based on soil properties. The water flux combines the upstream total area ($A$) and local precipitation ($P$)[83]. Finally, the weathering impact of precipitation and its role in river incision enhancement is incorporated by scaling the erodibility with local precipitation rate and $d$ (a positive exponent estimated from field-based relationships[84] as 0.42).

### Representation of Sahul physiographic and climatic conditions from 75 to 35 ka

To examine the role physiography played in the peopling of Sahul after 75 ka, we use two main inputs in our model. First, we take advantage of a recent high-resolution reconstructed digital elevation model of the region at similar age[9] (Fig. 1a). In this model (clipped between 3 and 45° S, 111–155° E), New Guinea, mainland Australia and Tasmania form a single continent by considering an initial coastline at the −68 m isobath around 75 ka[10,85] that decreases to −85 m around 65 ka before increasing by ~25 m over the remaining 25 kyr[30]. A series of existing regional- and global-scale digital elevation models (GEBCO grid, ausbathytopo grid, SRTM datasets, gbr100)[9] was used to generate this digital product with a resolution of 0.0025° (~250 m). We then resample this initial paleo-elevation to a coarser resolution of 0.01° (~1 km) corresponding to a grid of more than 19 million cells. This coarsening step results in the smoothing of some landscape features but was required because the highest resolution was too large to be computationally tractable over the different modelling approaches that are performed in this study (i.e., physiography, current density, and mechanistic movement modelling).

Second, to simulate riverine processes, a paleo-precipitation dataset needs to be provided to goSPL. Here, we chose the one generated using a variant of the coupled atmosphere-ocean-vegetation Hadley Centre model and named HadCM3[36]. This fully coupled ocean–atmosphere global circulation model covers the last 120 kyr at 2 kyr temporal resolution. Its land surface algorithm accounts for the representation of soil moisture and terrestrial evaporation (considering temperature, vapor pressure, and $CO_2$ concentration)[36]. The resolution of the atmospheric model is 2.5° in latitude by 3.75° in longitude and predicted paleo-precipitations were interpolated to our paleo-elevation resolution. We use each 2-kyr precipitation map as initial rainfall condition in goSPL (corresponding mean precipitation distribution for the simulated time is presented in Fig. 1b). Relatively arid conditions dominate marine isotope stage (MIS) 3 in Sahul from 59 to 24 kyr[34,86]. From the predicted paleo-precipitation dataset[36], we extract for each archaeological locations[9] (Supplementary Table 1) the corresponding rainfall anomalies for the simulated period (Fig. 1c) and find relatively small variabilities across the different sites (mostly related to the model resolution and ranging between −400 and 300 mm/yr). At local scale, most curves exhibit drier conditions during the first 10 kyr, wetter climates from 65 to 55 kyr, followed by drier distribution after 40 ka. Similarly, evaluation of Australian hydroclimate during MIS3 using multi-proxy metadata analysis and ensemble-hindcasting approaches has shown no major shift in continental-scale climate regime between 75 and 35 ka, with precipitation anomaly decreasing by less than 0.5 mm per day over that period[86]. At regional scale, an analysis based on 40 Australian MIS 3 records[34] suggests spatially variable climates from 75 to 50 ka, followed by wetter conditions up to 40 ka, then becoming drier until 25 ka.

From the resampled digital elevation model, paleo-precipitation maps, and the relative coastal position, we generate the input files for goSPL and run a forward simulation from 75 to 35 ka with outputs generated every 1 kyr. The simulation is launched over 80 CPUs and displays the impacts of Earth surface processes (i.e., riverine and hillslope) and records the evolving physiographic characteristics (paleo-stream paths, paleo-lakes, and associated paleo-drainage basins evolution) as well as the associated continental erosion, transport, and deposition. Calibrated model parameters (Eq. 1) predict denudation rates for the Great Australian Escarpment averaging 17 mm/kyr (max. rate of 82 mm/kyr – Supplementary Fig. 1c) similar to those found in other post-orogenic landscapes and passive margin settings[87] and comparable to those published from cosmogenic radionuclides analysis for the region[87,88] (i.e., [10]Be-based basin-wide denudations rates). Similar rates are also predicted over the Tasmanian landscape[88]

(max. ~58 mm/kyr); while the northern part of Sahul, along the Papua New Guinea orogen (Supplementary Fig. 1b), exhibits much higher denudation rates of up to 300 mm/kyr due to the combined effects of higher precipitations and steep topographic gradients[89]. As expected, smaller denudation values, and even endorheic sedimentation, are predicted in central and western Australia owing to their relatively flat landscapes and overall low rainfall regimes (Supplementary Fig. 1d), except for high relief regions (e.g., the Kimberley Plateau where the model predicts a maximum denudation rate of ~41 mm/kyr).

## Morphometrics characterisation of Sahul physiography

First, we calculate the local relief from predicted paleo-elevation and normalise it to derive a slope index (Supplementary Fig. 2). We used the slope instead of defining an index purely based on elevational range as movement over relatively flat elevated landforms (e.g., plateaus) might be easier than over deeply dissected low-elevation regions (e.g., incised valleys or canyons). Then we define an environmental index (Supplementary Fig. 2) that considers endorheic lakes and oceans as dispersal barriers and assume that major rivers (>80 m³/s) also represent strong barriers to crossing[38] (as an example the Murray-Darling, Flinders, and Murrumbidgee rivers have discharge rates of 767, 122 and 120 m³/s respectively). However, we do not deem these major rivers as fully impermeable in the cost calculation and allow their normalised costs to vary linearly with predicted flow rates from goSPL between 0.7 and 0.95. For more modest rivers, their costs to cross scale with their predicted flow rates. Costs along rivers are assumed to be small (<0.25), facilitating movement down or upstream accounting for the greater access to resources. In addition, due to their lack of water and food, the environment index defines high cost (>0.9) in the arid and flat regions identified from a combination of topography index, paleoclimate precipitation maps[34] and net primary productivity[7,26] (e.g., around Lake Eyre). Finally, we define a physiographic diversity metric that quantifies paleo-landforms. It uses the topographic position index on each cell $i$ ($TPI_i$) that measures the relative relief[90]:

$$TPI_i = z_i - \sum_{k=1}^{n} z_k/n \quad TPI_S = 100 \times (TPI - \overline{TPI})/\sigma_{TPI} \quad (2)$$

where $z_i$ is the considered elevation at cell $i$ and is subtracted to the mean of its surrounding cells ($z_k$) with $n$ is the number of cells contained inside an annulus neighbourhood. Topographic position is an inherently scale-dependent calculation[91]. To circumvent this problem, $TPI$ is computed considering 2 scales of observation, a fine one ranging between 0.02° and 0.05° and a coarser one from 0.1° to 0.15°. Because elevation is generally spatially autocorrelated, $TPI$ values increases with scale, making it difficult to compare both scales of observation directly. To overcome this issue, we calculate a standardised $TPI_S$ (Eq. 2), where $\overline{TPI}$ is the mean over the entire grid and $\sigma_{TPI}$ its standard deviation[92]. From the continuous variable, we then derive categorical variables by defining 7 categories from the slope, 5 from the water flux, and 10 from the $TPI_S$[93] (Supplementary Table 2). The physiographic diversity index $P_{DIV}$ (Supplementary Fig. 2) is finally obtained from Shannon's equitability (continuous variable [0,1]), which is calculated by normalising the Shannon–Weaver diversity index ($d_{SW}$):

$$d_{SW} = -\sum_{k=1}^{C} p_k \ln(p_k) \quad P_{DIV} = d_{SW}/\ln(C) \quad (3)$$

with $p_k$ the proportion of observations of type $k$ in each neighbourhood and $C$ the number of categorical variables (here $C = 3$). By taking, for each pixel, the maximal cost from the different morphometrics, we obtain a final physiography-dependent resistance map (Supplementary Fig. 3) that will then be used to compute human dispersal across Sahul landmass.

## Modelling Sahul dispersal

We use the mechanistic model SiMRiv[24] that simulates spatially explicit stochastic movements (multi-state Markov model[94]) accounting for landscape heterogeneity and perceptual range (i.e., the radius up to which an individual perceives its surroundings). To run the mechanistic movements simulation, we rely on the normalised resistance maps of Sahul physiography (Supplementary Fig. 3 – defining the underlying environmental complexity of the region) and we set an initial entry point. Several most likely dispersal routes and migration hypotheses through Wallacea have been inferred either from biogeographic connectivity[10,27], archaeological data[1,9,33], or genetic and genomic evidences[95–97]. While still speculative, two main pathways for the oldest peopling phase have been proposed. The southern route passes through the island of Timor and suggests an entry into Sahul around 75 ka *via* the Timor Sea shelf (northwest Australia) that was partially exposed at that time[2,10,27,31]. The northern route enters Sahul around 73 ka from Borneo to Misool in northwest Papua then into the Bird's Head (Vogelkopf Peninsula)[8,10,27]. Here we chose this double entry scenarios[2,8] for our entry points and run two sets of mechanistic simulations (Supplementary Fig. 4).

Each simulation starts from one of the entry points with a single walker allowed to move based on normalised resistance maps defined at 2 km resolution (Supplementary Fig. 3). Displacements across the landmass are simulated with a two-state movement (Lévy-like walker[45]) that alternates between random and correlated random walks[44] where the incremental direction is correlated with the prior one (turning angle concentration set to 0.95). To inform the mechanistic model on the probabilities of changing between the two states, SiMRiv provides a transition matrix which is defined here to favour correlated walks (state switching probabilities of 0.01 to change from random to correlated walks and of 0.002 to change from correlated to random walks). We also define a step length (maximum distance per step) for both states of 1 km to avoid unrealistic jumps over high-cost surfaces. Finally, we set a perceptual range to 10 km: the movement decisions at each step accounts for the resistance that the walker *sees* in a 10 km radius around the current location (i.e., the walker attempts to minimise resistance when moving within this range)[23].

SiMRiv does not incorporate time explicitly but uses a maximum number of steps (i.e., displacements) that a walker is allowed to make. Here this number is set to 10 million for the simulated period ranging from 75 to 35 ka. This corresponds to 250 Lévy walks of up to 1 km per year. It adds up to an upper limit of 250 km travelled per year. From reported fast residential mobility of up to 50 km in length found in African hunter-gatherers and Australian Aboriginal people[8,52,54], this upper limit would represent a maximum of 5 residential moves per year. To account for changes in environmental conditions during migration, resistance maps are updated every 1 kyr (~250,000 walker steps) with Lévy walkers integrating the change in physiography as they continue their journey across Sahul landmass.

## Reporting summary

Further information on research design is available in the Nature Portfolio Reporting Summary linked to this article.

## Data availability

The high resolution reconstructed digital elevation model of the region is available as shape files from the Supplementary Materials in Crabtree et al. [9]. Paleo-precipitation maps from the HadCM3 coupled atmosphere-ocean-vegetation Hadley Centre climate model[36] are available from the Bristol Research Initiative for the Dynamic Global Environment (BRIDGE) website: https://www.paleo.bristol.ac.uk/ummodel/users/Singarayer_and_Valdes_2010/new2/. Hindcasted net primary production values for Sahul[7] produced by the LOVECLIM global circulation model[26] are available at 1 kyr resolution from https://doi.org/10.5281/zenodo.4453767. Source codes for the generation of

the figures and associated data processing have been deposited along with the raw data in the same Zenodo repository (https://doi.org/10.5281/zenodo.10889086).

## Code availability

The scientific software used in this study, goSPL[25], is available from https://github.com/Geodels/gospl and https://doi.org/10.5281/zenodo.523461. The software documentation can be found at https://gospl.readthedocs.io. SiMRiv[24] is available at https://github.com/miguel-porto/SiMRiv. The open-source python interface for the Generic Mapping Tools (https://www.pygmt.org) was used for two-dimensional map visualisation except for the ones in Supplementary Figs. 1 and 8 that were created with the open-source Paraview software (https://www.paraview.org).

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

## Acknowledgements

T.S. is supported by the Australian Research Council DARE Centre (project number IC190100031). R.J.-B. is supported by an Australian Research Council Discovery Project (project number DP220100195). I.M. is the recipient of an Australian Research Council Future Fellowship (project number FT220100184). This research was also undertaken with the assistance of resources from the National Computational Infrastructure (NCI), which is supported by the Australian Government, and from Artemis HPC Grand Challenge supported by Sydney Informatics Hub at the University of Sydney.

## Author contributions

T.S. and R.J.-B. conceived the study, T.S. conducted the numerical experiments, L.H., M.L. and T.S. worked on the physiography index definition and interpretations. T.S., I.M., L.H. and R.J.-B. analysed the mechanistic movement simulations. T.S. wrote the initial draft and L.H., I.M., R.J.-B. and M.L. reviewed the manuscript.

## Competing interests

The authors declare no competing interests.
