## [Peer Review File · Nature Communications]

Physiography, foraging mobility, and the first peopling of SahulREVIEWER COMMENTS

Reviewer #1 (Remarks to the Author):

My recommendation is for acceptance after revisions, particularly to portions of the methods.

What are the noteworthy results?

In my view the most noteworthy are the time of arrival speed of migration to different sites as those are the first estimates of these values done with a Levy walk type of model.

Will the work be of significance to the field and related fields?

Yes, both for those interested in the archaeology of Sahul, early migrations in general and those interested in modeling population displacement over land.

How does it compare to the established literature? If the work is not original, please provide relevant references.

I believe the work is original.

Does the work support the conclusions and claims, or is additional evidence needed?

It does a good job in supporting its conclusions

Are there any flaws in the data analysis, interpretation and conclusions? Do these prohibit publication or require revision?

Not that I could note

Is the methodology sound? Does the work meet the expected standards in your field?

Yes, although I believe the portions of the text detailing the methodology should be improved for details and clarity (this is in fact my only and main concern)

Is there enough detail provided in the methods for the work to be reproduced?

No.

I have made several comments to the uploaded annotated pdf copy of the paper and these are an integral part of my review

**Please note that the attachment to Reviewer #1's report can be found here:

<https://drive.google.com/drive/folders/19-dv2NlxIV4b45YmoNuKXbZg-2an9cw8?usp=sharing>

Reviewer #2 (Remarks to the Author):

I am not a strong believer in lengthy peer reviews, so I will keep my comments brief.

I recognize the significance of what the authors have done, introducing a "new" kind of modeling to the discussion of the peopling of Sahul that appears to complement other recent approaches. I think it is a valuable contribution that should be published, but not in its current form. They have more work to do.

My specific constructive feedback:

1) You can't acknowledge the need to factor in non-environmentally-deterministic variables (e.g., social, cultural, demographic), as other recent approaches successfully have, and then completely

ignore them. Repeating a statement that you acknowledge them and will be ignoring them, across the manuscript, does not absolve you of the need to factor them in. It is a significant missing component of your model and its absence initially strikes me as one tied to inconvenience, inexperience, or both. If you factor them in, then do a comparison to a version of your model that doesn't, and there is little to no difference in the results, then and only then can you say with confidence that they don't matter. Dismissing them out of hand is insufficient.

2) You mention that your model is complementary to other recent approaches, yet you do not directly compare the results of your model to the results of those approaches, apart from rough arrival time estimates. You should be able to overlay your spatiotemporal maps on top of those from the other approaches and obtain more meaningful results with respect to where your model agrees with or deviates from the others. Saying "our average speed is X and theirs is Y" is insufficient and represents a large missed opportunity to highlight in a more convincing manner that what you have developed is significant and complementary.

3) Your employment of least cost path modeling is surprisingly simplistic. It's as if you're comparing a Formula One race car (your complex modeling framework) to a tricycle (executing a basic 64-year-old graph search function in scikit-image). For the purposes of your submission, it's effectively a straw man that is not at all convincing to someone who actually understands how least cost modeling works and what the state of the art is--especially when it comes to modeling travel with limited or no knowledge of specific origins and/or destinations. I recommend that you either remove the comparison from your submission or work with a specialist in Geographic Information Science (GIS) to create something of commensurate sophistication to your model, after which a comparison makes sense.

4) You provide a moderately convincing rationale for selecting a single entry point for your model, but then you go on to say that your results are diffuse enough that you weren't able to discern any clear spatial travel patterns beyond some preference for rivers. If you modeled more than one entry point, which is highly defensible, and combined the outputs, I'm betting you'd start to see patterns. You in essence have one very large and complex data point right now. You need more before you can convincingly say that there are no patterns. Many of the other recent approaches use multiple coastline entry points across the continent and patterns emerged that had to be evaluated and discussed.

5) Generating archaeological site likelihood estimates for 5000 randomly generated points is an odd and largely useless way to demonstrate the potential of your model for prospection. You should run the calculations for every land cell in your grid to produce a "heat map." Archaeologists have done this within GIS environments for at least the past twenty years. If running every cell through the process is computationally cost-prohibitive, although I'm not sure why that would be the case, a compromise would be to do every other cell (or slightly coarser, like every third or fifth) and then interpolate between the results to produce the map.

Reviewer #3 (Remarks to the Author):

It is difficult to easily profile the noteworthy results in the paper. As the authors state this modelling using Lévy walk foraging patterns produces speed of settlement and radiation patterns which are consistent with (most recently) the stochastic demographic models of Bradshaw, C.J. A., Norman, K., Ulm, S.G., Williams, A.N., Clarkson, C., Chadoeuf, J., Lin, S.C., Jacobs, Z., Roberts, R.G., Bird, M.I., Weyrich, L.S., Haberle, S.G., O'Connor, S., Llamas, B., Cohen, T.J., Friedrich, T., Veth, P., Leavesley, M., and F. Saltré 2021 Stochastic models support rapid early peopling of Late Pleistocene Sahul. *Nature Communications* 12(1): 2440-2422. This is not surprising given the same archaeology data sets referenced in Crabtree et al. 2021 are relied on. The following of riverine corridors by people into the interior is, again as they note, not a new idea however the early roles of lakes in this radiation and

better controls for landscape evolution are well developed. The early use of riverine corridors is discussed by Wallis and Hiscock 2005, Smith 2013 and Veth, P. et al 2022 *Beyond the Barriers: A New Model for the Settlement of Australian Deserts* in I. J. McNiven and B. David (eds) *The Oxford Handbook of the Archaeology of Indigenous Australia and New Guinea* DOI: 10.1093/oxfordhb/9780190095611.013.32. Indeed it is a tenet of the 'desert transformation model' where people adapt to arid conditions in situ - see Hiscock, P. & Wallis, L. A. Pleistocene settlement of deserts from an Australian perspective. *Desert peoples: Archaeological 563 perspectives* 34–57 (2005).

The work is of some limited significance in that it deploys a new methodology. However, the significance of the highly prospective sites with archaeological potential (Figure 5 and locations A to J) guiding future work requires much more justification and expansion. This can't simply be cast as new and interesting ways to think about the archaeology. The geomorphic settings are so radically different between these locales it seems unsatisfactory to simply note there will be taphonomic differences in site preservation between them. For example, what specific attributes of Locale C on the Nullagine River make it so prospective other than a higher likelihood of having been visited earlier and more likely to have been used repeatedly? The regional archaeological context appears to be missing and this because the regional syntheses which highlight outstanding archaeological questions (e.g. Jankowski and Stern's recent work on Willandra Lakes and Smith's 2013 *Desert Volume*) are some of the obvious and missing sources. Big data models using paleoclimate proxies will make it hard to drill down into regional differences, however there are such major variations in the uncoordinated vs coordinated drainage systems of (say) the Western Desert vs Lake Eyre Basin and Willandra Lakes as to merit at least a brief discussion.

One major weakness in the paper is that it hardly considers Pleistocene coastal environments, despite the fact that the founding Australian populations were maritime-capable people (and see recent synthesis papers from Wallacea by O'Connor, Keely and Shipton). Instead, it seems the authors have dismissed the coastal evidence as being "peripheral", in line with what O'Connell and Allen suggested in decade-old publications. The authors should at least explore the Pleistocene coast and people's movement into those environments - such as the Veth et al. 2017 QSR Boodie Cave paper and the Pleistocene coasts 2022 QSR paper by Ditchfield et al. which make a persuasive case that the Pleistocene coast was both productive and occupied. This does not seem to have been considered by the authors. These papers, and others, also explain why some of the archaeology in Pleistocene coastal sites can appear 'peripheral' since most of the actual sites that occurred within a few km of nearly all Pleistocene coasts are drowned. There are however, records of Pleistocene shellfish collection, high densities of lithics and terrestrial fauna from 50 - 42 ka deposits on the North West Shelf. The authors seem to have used old coastal ideas and literature, such as Bowdler's coastal colonisation model. Ultimately, by ignoring the previous papers and data, the paper falls into the trap of continuing to perpetuate overly-terrestrial and biased narrative of early Pleistocene Australia.

Otherwise the methodology appears sound, including landscape evolution, human mobility models and normalised cost mapping. There is an apparent contradiction between the case for landscape evolution, however, and the resilience and locations of continental water sources remaining static for over 65,000 years - see Bird, M. I., O'Grady, D. & Ulm, S. *Humans, water, and the colonization of Australia*. *Proc. Natl. Acad. Sci.* 113, 448 11477–11482, DOI: 10.1073/pnas.1608470113 (2016). There are also relevant palaeoclimate models for the lake/riverine entry hypothesis in the north which should be cited. For example;

De Deckker, P., M. Moros, K. Perner, T. Blanz, L. Wacker, R. Schneider, T.T. Barrows, T. O'Loingsigh and E. Jansen 2020 *Climatic evolution in the Australian region over the last 94 ka - spanning human occupancy - and unveiling the Last Glacial Maximum* *Quaternary Science Reviews* 249 (2020): 106593

Cadd H et al (2021). *A continental perspective on the timing of environmental change during the last glacial stage in Australia*. *Quaternary Research* 102,

5-23. <https://doi.org/10.1017/qua.2021.16>

The paper has many typographic and referencing issues. e.g. Rhys Jones 1979 becomes Rhys, J. and so on. The grammatical errors could largely be removed if the text were carefully proofed by author(s) speaking English as a first language.

The paper has merit and will be of interest to Australian and broader archaeologists, however needs further work in the areas identified above.

Responses to reviewers' comments on the initial version of our manuscript.

We were very pleased with the outcomes of the review process, both reviewers acknowledged the significance of our work, its valuable contribution to the discussion of the peopling of Sahul as well as the novelty of the approach that is proposed here based on Lévy walk modelling. Yet, the reviewers raised several points needing additional work and explanations, and we would like to thank them for their thoughtful comments and efforts towards improving our manuscript.

General notes on revised manuscript

The main changes in this revised version are summarized here:

- 1- We redefined our Sahul resistance maps (used to evaluate human dispersal with the mechanistic model) to account for net primary productivity (extracted from LOVECLIM climate model as described in Bradshaw et al., 2019) also forcing low resistance to movement along Sahul coastline (up to 50 km inland) to provide more opportunities for coastal colonisation (Veth et al., 2017; Ditchfield et al., 2022). We then rerun our mechanistic model from 75 to 35 ka using these new resistance maps.
- 2- We propose a new set of simulations with a southern entry point into modern-day Kimberley region via Timor following the double entry scenarios proposed by Kealy et al. (2018) and Bradshaw et al. (2023). Similar to what was already done for the northern entry, we perform a probabilistic analysis of the migration patterns, residential likelihood and travelled distance and speed for both entry points based on archaeological sites identified in Crabtree et al. (2021).
- 3- We provide a heat map of probability of presence of human across Sahul's landmass by aggregating the two set of simulations (northern and southern entry points) and counting the number of *walkers* presents on each cell (0.05° grid resolution). This probability map is then compared to the superhighways predicted with the least-cost pedestrian travel model of Crabtree et al. (2021).

In the following, we respond to each specific comment. We answer suggestions of each reviewer and discussed the modifications that were made to address them.

REVIEWER 1

Note: reviewer have made several comments to the uploaded annotated pdf copy of the paper.

This passage of the text is a bit harsh with LCPs. Least cost experiments could be set up with with a general bearing, such as "towards the east", or and this in my view is a general direction of movement not "a planned destination". Also, LCPs do not require "complete knowledge of the environment", just of the areas adjacent to agent's position. In fact, this is an aspect shared by the model adopted by authors here. In the method section: Don't generalize. These are assumptions peculiar to this and many others, but not all, LCP experiments.

We note that a similar point was raised by reviewer 2. Here, we were mostly referring to early LCP models (like the one presented in the Method section of our initial manuscript) that mainly work on the assumption that the whole world is perfectly known, which does not represent human mobility accurately in all cases (Gravel-Miguel and Wren, 2018). We agree that most recent studies are using improved versions of LCPs such as agent-based least-cost path (AB-LCP) that we discuss in the last sentences of this first paragraph (lines 28 to 31). To avoid any confusion, we have decided to remove this sentence and also the section discussing the simplest LCP approach presented in the Method section.

This might be a good place to remind the reader that these estimates were made on populations that have existed for a long time on the observed area, and also that those areas are within regions already occupied by other human groups. That

this makes them not perfectly analogous to the problem at hand and that foraging strategies adopted by groups entering a human-free and completely unknown area might be different. I don't mean here to invalidate the authors choice of Lévy walk, which I consider quite reasonable for the problem at hand.

We appreciate this comment, and we modified the sentence accordingly (addition in blue):

While these latter studies have focussed on areas within regions already occupied by other human groups, Lévy walk strategies are likely a key component of hominins foraging movements since the adoption of a hunter-gatherer lifestyle nearly 2 million years ago in human ancestors, playing an important role in human mobility and in our capacity to exploit new environments rapidly and efficiently.

The use of the term "main landscape features" here could be confusing for some readers who would consider parameters such as vegetation cover a main feature of the landscape.

We have replaced the term by **main geomorphic features**.

Nothing wrong with this sentence, but I believe that somewhere in this section authors should call attention to the lack of some estimate of, to be generic, an npp-like parameter in their resistance surface. Their (in my view justified) adoption of Levy walk based on foraging behavior and foraging behavior is dependent on npp (or vegetation cover or carrying capacity...). In the same section: This reinforces the comment above about npp's importance to displacement rates (Arnhem's npp >> western desert's). And further down in the manuscript: Sorry to bang on this drum again, but spatially explicit changes to this parameter could be a nice way to incorporate npp/carrying capacity in your simulations. Just a comment.

Following reviewer comments and as mentioned in the general notes, we are now incorporating maps of Net Primary Productivity (similar to Bradshaw et al., 2019) as an additional index in our resistance maps calculation. The corresponding text in this section now reads: *In addition, a regional net primary productivity layer is defined from LOVECLIM climate reconstruction as an indicator of relative ecological carrying capacity of the ecosystem (Extended Data Fig. 2) and low cost areas are enforced along coastal regions.*

We have also modified Extended Data Fig. 2 and 3 accordingly.

Are both random and correlated steps considered or only the correlated ones?

Based on reviewer comment we have modified the sentence which now reads: *To estimate the number of residential moves out of the Lévy-walk simulations, we aggregate successive displacements (combining random and correlated steps) made within a specified radius and consider them as part of a single mobility event in their migration.*

This passage can lead some to incorrectly understand that other values between 25 and 50 km were used.

This has been modified accordingly in the revised version: *two distances set at 25 and 50 km respectively.*

Is a step not a displacement unit?

This has been modified to: *the number of random and correlated walks between residential moves.*

Does a single realization track the position of a single group starting from a single point in space? I believe that is the case, but don't see this clearly stated in the text. Are the number of walkers here a sum over all realizations or the same group walking into a certain area many times?

We appreciate this comment and have modified the text accordingly: *For each realisation (that tracks the position of a single group of individuals starting from one of the two entry points), we evaluate (1) the likelihood...* The number of walkers for each realisation (i.e., mechanistic simulation) is now 10 million and 5,000 realisations are performed for each entry point.

Are you sure you are not off by an order of magnitude here? At least the figure puts most 25 km radius min distances above 10^4 , and I think the 50 km text values also don't match the fig. super well...

Thank you for pointing out this inconsistency between the graph and the distances that were provided in the text. We have now modified the text according to the new distances calculated for the two entry points using the new resistance maps and estimated distances in the text match the ones presented in Fig. 5.

Not surprising, given the model's resolution...

We now acknowledge that this is likely due to the paleo-climate resolution and made changes to the corresponding sentence: *find relatively small variabilities across the different sites (mostly related to the model resolution and ranging between -400 and 300 mm/yr).*

Minor: While the Methods "Physiography modelling" section mentions the "multiple flow direction algorithm" it might be useful to, somewhere around here, inform the interested but non-specialist reader that flow rates come from goSPL. Some more information on how the location and size of arid regions was determined would be useful. Were paleoclimate model or proxies used or are arid areas estimated from topography alone?

As suggested by the reviewer, we have modified this section of the Methods to reflect that flow rates are derived from the landscape evolution model and to explain the proxies used for arid areas estimation: *...with predicted flow rates from goSPL between 0.7 and 0.95. ... identified from a combination of topography index, paleoclimate precipitation maps and net primary productivity.*

Detailing the methodology should be improved for details and clarity.

What is meant by this? Movement is quantized in 200m "steps" with direction of movement of some number of consecutive steps being completely random followed by a number of consecutive steps where direction at time step n is correlated to direction at step n-1? What does the transition value matrix mean in terms of the average fraction between random and correlated steps? Given Nat Comm's wide range of reader competency a bit more non-specialist language might be welcome here. What varies between each realization and how? What do 5 million steps mean? Does the simulation stop after 5 million steps are calculated? It is not clear to me how the evolution of physiographic features estimated by goSPL was used to update resistance surfaces.

We have completely rewritten this section of the Methods with reviewer's comments and suggestions in mind: section on page 16 of the revised manuscript "Modelling Sahul dispersal". First, we remove the part related to LCPs (see answer to first comment), we also discuss the double entry points specifying the initialisation of each realisation with a single walker that we then track over the 40 kyr of simulation adjusting the path to temporal changes in resistance maps every 1 kyr. In addition, we explain what the number of walker steps mean and how it was tuned based on the modelled timeframe and to fit the purpose of our study. Finally, we have avoided, when possible, the use of too technical specialist language.

Does this mean "walker attempts to minimize resistance when moving within this range"? Non-native speaker here, but cannot bypass also convey the idea that high resistance cells are "jumped over"?

We are now using the wording proposed by the reviewer instead of "bypass".

REVIEWER 2

You can't acknowledge the need to factor in non-environmentally-deterministic variables (e.g., social, cultural, demographic), as other recent approaches successfully have, and then completely ignore them. Repeating a statement that you acknowledge them and will be ignoring them, across the manuscript, does not absolve you of the need to factor them in. It is a significant missing component of your model, and its absence initially strikes me as one tied to inconvenience, inexperience, or both. If you factor them in, then do a comparison to a version of your model that doesn't, and there is little to no difference in the results, then and only then can you say with confidence that they don't matter. Dismissing them out of hand is insufficient.

We mention twice (in the discussion and the conclusion) that we do not consider social, cultural, and economic aspects because this is indeed one of the limitations of our approach. However, we never suggested that they did not matter. Saying that, we are unaware of any recent numerical models able to factor all these different aspects and the references that we provided do not correspond to models that actually do so. As a matter of fact, this limitation is inherent to all the models (even the most recent ones, e.g., Bird et al., 2016; Bradshaw et al. 2019; Bradshaw et al., 2021; Crabtree et al., 2021; Bradshaw et al., 2023) that have been published on Sahul migration and likely other regions in the world. As explained by some of these authors:

Crabtree et al., 2021 – Nat. Human Behaviour: *A limitation of this study is that our methods intentionally avoid considering many types of potential biases, including concerns of demographic composition or traveller velocity, the types of eco-systems encountered and, perhaps most controversially, cultural influence.*

Bradshaw et al., 2021 – Nat. Communications: *Our model provides a baseline approximation for the relative timing and spread of human dispersal in Sahul on the basis of ecological parameters alone. The reliance here on ecological principles explicitly does not mean that human dispersal in Sahul did not involve deliberate social, cultural and/or economic decisions.*

One point to note however is that the approach that we propose here (based on Lévy walker) has been shown to accurately represent contemporary hunter-gatherer populations (discussed in the second paragraph of the introduction), populations whose behaviours and decision-making processes are embedded with cultural knowledge and practices (we have added this sentence in our revised conclusion). Combining our approach more explicitly with social and cultural aspects (either as part of an additional model or from existing dataset) would represent a considerable advance.

You mention that your model is complementary to other recent approaches, yet you do not directly compare the results of your model to the results of those approaches, apart from rough arrival time estimates. You should be able to overlay your spatiotemporal maps on top of those from the other approaches and obtain more meaningful results with respect to where your model agrees with or deviates from the others. Saying "our average speed is X and theirs is Y" is insufficient and represents a large, missed opportunity to highlight in a more convincing manner that what you have developed is significant and complementary.

Following reviewer's comment, we are now comparing our model with the predicted superhighways from Crabtree et al. (2021) highlighted in Fig. 6 and discussed the similarities and differences between the 2 predictions in the last paragraph of the first section of the Discussion: "Predicted peopling of Sahul". We have chosen to compare the predicted migration pathways and corridors instead of speeds because our calculation of the speed of propagation is mostly obtained *a posteriori* in our approach and only provide a possible range that we compare to ranges proposed by other.

Indeed, and inherent to the Lévy walk approach, we do not explicitly incorporate a walking speed in our approach like the Tobler's hiking function (Bradshaw et al., 2023). As a result, velocity estimates (as explained in the manuscript) are obtained assuming only two entry points and considering only two entry times. We acknowledge the limitations of this approach too: the peopling of Sahul was likely accomplished in successive waves unrestricted to the chosen northern and southern entries nor to a 75-73 ka double-entry scenario. Also, the dates from the considered archaeological sites do not necessarily represent the earliest presence of people at a site and the sites may not represent the first presence of people in a region.

Your employment of least cost path modeling is surprisingly simplistic. It's as if you're comparing a Formula One race car (your complex modeling framework) to a tricycle (executing a basic 64-year-old graph search function in scikit-image). For the purposes of your submission, it's effectively a straw man that is not at all convincing to someone who actually understands how least cost modeling works and what the state of the art is--especially when it comes to modeling travel with limited or no knowledge of specific origins and/or destinations. I recommend that you either remove the comparison from your submission or work with a specialist in Geographic Information Science (GIS) to create something of commensurate sophistication to your model, after which a comparison makes sense.

We note that a similar comment was made by reviewer 1. Here, we were mostly referring to early LCP models (like the one presented in the Method section of our initial manuscript) that mainly work on the assumption that the whole world is perfectly known, which does not represent human mobility accurately in all cases (Gravel-Miguel and Wren, 2018). We agree that most recent studies are using improved versions of LCPs such as agent-based least-cost path (AB-LCP) that we discuss in the last sentences of this first paragraph of the introduction (lines 28 to 31). To avoid any confusion and following reviewer's advice, we have decided to remove this sentence and also the section discussing the simplest LCP approach presented in the Method section.

You provide a moderately convincing rationale for selecting a single-entry point for your model, but then you go on to say that your results are diffuse enough that you were not able to discern any clear spatial travel patterns beyond some preference for rivers. If you modelled more than one entry point, which is highly defensible, and combined the outputs, I'm betting you'd start to see patterns. You in essence have one very large and complex data point right now. You need more before you can convincingly say that there are no patterns. Many of the other recent approaches use multiple coastline entry points across the continent and patterns emerged that had to be evaluated and discussed.

We agree with the reviewer. As explained in the general notes of the revised manuscript, we are now proposing a new set of simulations with a southern entry point into modern-day Kimberley region via Timor at 75 ka following the double entry scenarios proposed by Kealy et al. (2018) and Bradshaw et al. (2023). Similar to what was already done for the northern entry, we perform a probabilistic analysis of the migration patterns, residential likelihood and travelled distance and speed for both entry points based on archaeological sites identified in Crabtree et al. (2021). We also combined the outputs of our two set of simulations to evaluate the possible presence of migration pathways that we then compare to the superhighways model proposed by Crabtree et al. (2023) and also used in Bradshaw et al. (2023).

Generating archaeological site likelihood estimates for 5000 randomly generated points is an odd and largely useless way to demonstrate the potential of your model for prospection. You should run the calculations for every land cell in your grid to produce a "heat map." Archaeologists have done this within GIS environments for at least the past twenty years. If running every cell through the process is computationally cost-prohibitive, although I'm not sure why that would be the case, a compromise would be to do every other cell (or slightly coarser, like every third or fifth) and then interpolate between the results to produce the map.

As explained above and following the reviewer suggestion, we have calculated a heat map of probability of presence of human across Sahul's landmass by aggregating the two set of simulations (northern and southern entry points) and counting the number of *walkers* presents on each cell (0.05° grid resolution). We decided to keep the evaluation of the 5,000 generated points as well in our manuscript as it was also considered of interest by reviewer 1. While the calculation of the probability of presence of humans across Sahul is not too prohibitive (we still must aggregate 10^{11} geolocations onto a 0.05° grid), the approach used to compute number of visits to a specific site required to track and cluster each walker along their trajectory for each realisation before grouping them for the entire set of experiments. This type of calculation would be too computationally expensive (especially in terms of memory) for a continental exploration across Sahul at 0.05°. Instead, we envision that this approach would be useful and applicable when willing to discriminate between possible archaeological targets in a region of interest.

REVIEWER 3

The paper has merit and will be of interest to Australian and broader archaeologists, however, needs further work in the areas identified above. As the authors state this modelling using Lévy walk foraging patterns produces speed of settlement and radiation patterns which are consistent with (most recently) the stochastic demographic models of Bradshaw et al. (2021). This is not surprising given the same archaeology data sets referenced in Crabtree et al. (2021) are relied on. The following of riverine corridors by people into the interior is, again as they note, not a new idea however the early roles of lakes in this radiation and better controls for landscape evolution are well developed. The early use of riverine corridors is discussed by Hiscock & Wallis (2005), Smith (2013) and Veth et al. (2022). Indeed, it is a tenet of the 'desert transformation model' where people adapt to arid conditions in situ - see Hiscock & Wallis (2005).

First, we thank the reviewer for the additional references that we have all added to our manuscript. We think that the fact that our study produces similar speeds and radiation patterns consistent with other techniques is noteworthy not only because it increases our confidence on the interpretations that were previously made (Crabtree et al., 2021; Bradshaw et al., 2021; Bradshaw et al., 2023) but also because it provides a new approach that could be used to complement existing ones. There are also specificities relative to each individual techniques that would make them more appropriate for particular application, hence making them complementary to each other. As an example, our computation does not rely on archaeological sites (which are only used *a posteriori* to infer migration speed), it makes it a great prospective tool for investigating regions with sparse data or no dating.

The work is of some limited significance in that it deploys a new methodology. However, the significance of the highly prospective sites with archaeological potential (Figure 5 and locations A to J) guiding future work requires much more justification and expansion. This can't simply be cast as new and interesting ways to think about the archaeology. The geomorphic settings are so radically different between these locales it seems unsatisfactory to simply note there will be taphonomic differences in site preservation between them. For example, what specific attributes of Locale C on the Nullagine River make it so prospective other than a higher likelihood of having been visited earlier and more likely to have been used repeatedly? The regional archaeological context appears to be missing and this because the regional syntheses which highlight outstanding archaeological questions (e.g. Jankowski and Stern's recent work on Willandra Lakes and Smith's 2013 Desert Volume) are some of the obvious and missing sources. Big data models using paleoclimate proxies will make it hard to drill down into regional differences, however there are such major variations in the uncoordinated vs coordinated drainage systems of (say) the Western Desert vs Lake Eyre Basin and Willandra Lakes as to merit at least a brief discussion.

Following reviewer's comment, we have modified the corresponding section better acknowledging available regional synthesis and additional approaches (e.g., local stratigraphic frameworks) that could be done to improve on our predicted archaeological sites.

...each of the selected areas (Extended Data Fig. 7), and thereby serve to infer the preservation potential at given locations that could then be further investigated with refined stratigraphic frameworks (i.e., combining sedimentological and soil micromorphological analysis^{1 50 79 81}).

We note that similarly to our predictions, many of the archaeological sites dated before 35 ka are located in highly variable geomorphological settings:

The archaeological database (Extended Data Tab. 3) equally shows significant environmental differences between locations with sites in the arid interior close to relict overflow systems (e.g., Willandra Lakes region⁷⁹), in sandy, shield and stony deserts characterised by internal and uncoordinated drainages and water holes (e.g., Western Desert East of the Pilbara uplands⁷³), or along confined sandstone gorges with abundant fluvial activity in lowland basins (e.g., Cranebrook Terrace⁸⁰).

But we also acknowledge the limitations inherent to the approach proposed here (1 km geomorphological resolution) with coarse paleo-precipitation reconstructions that hamper the ability of the model to inform at local scale (i.e. the scale used for sedimentological and soil micromorphological analysis) on the variations between uncoordinated and coordinated drainage systems:

Incorporating a refined temporal description of uncoordinated (e.g., the Western Desert⁷³) versus coordinated drainage systems (e.g., Lake Eyre Basin and Willandra Lakes⁷⁹), by integrating site specific variations inferred from local proxies^{36, 37, 39}, would further improve our predictions.

It is worth mentioning that these constraints exist for all approaches looking at continental scale evolution over geological timescale.

One major weakness in the paper is that it hardly considers Pleistocene coastal environments, despite the fact that the founding Australian populations were maritime-capable people (and see recent synthesis papers from Wallacea by O'Connor, Keely and Shipton). Instead, it seems the authors have dismissed the coastal evidence as being "peripheral", in line with what O'Connell and Allen suggested in decade-old publications. The authors should at least explore the Pleistocene coast and people's movement into those environments (such as the Veth et al., 2017; Ditchfield et al., 2022) which make a persuasive case that the Pleistocene coast was both productive and occupied. This does not seem to have been considered by the authors. These papers, and others, also explain why some of the archaeology in Pleistocene coastal sites can appear 'peripheral' since most of the actual sites that occurred within a few km of nearly all Pleistocene coasts are drowned. There are, however, records of Pleistocene shellfish collection, high densities of lithics and terrestrial fauna from 50-42 ka deposits on the North West Shelf. The authors seem to have used old coastal ideas and literature, such as Bowdler's coastal colonisation model. Ultimately, by ignoring the previous papers and data, the paper falls into the trap of continuing to perpetuate overly terrestrial and biased narrative of early Pleistocene Australia.

We have modified the study to account for this comment. As pointed out in the general notes at the top of this document, we have changed our Sahul resistance maps and assumed low resistance to movement along Sahul coastline (up to 50 km inland, see low-cost blue regions along the coast in Extended Data Fig. 3b and c) to provide more opportunities for coastal colonisation (Veth et al., 2017; Ditchfield et al., 2022). From our results and when considering the southern entry point, we do not think the archaeology of coastal sites would have been peripheral (i.e., our kernel density estimation shows relatively high values for those coastal sites as well as some of the highest percentages of walker presence - Fig. 3b and 4). When it comes to preferential pathways, we find moderate to high probabilities along the south-western coastlines as well as a likely migration corridor east of Lake Carpentaria that follows the eastern side of the Great Dividing Range similar to what was proposed in the superhighway model of Crabtree et al. (2021). What we suggest here is neither a fully continental/terrestrial migration nor an exclusive coastally restricted route but rather a wave of dispersal that would have radiated across the landmass following drainage basins and streams as well as coastlines (Fig. 6).

Otherwise, the methodology appears sound, including landscape evolution, human mobility models and normalised cost mapping. There is an apparent contradiction between the case for landscape evolution, however, and the resilience and locations of continental water sources remaining static for over 65,000 years (Bird et al., 2016). There are also relevant paleoclimate models for the lake/riverine entry hypothesis in the north which should be cited (De Deckker et al., 2020; Cadd et al., 2021).

We thank the reviewer for his comment and have added the reference to De Deckker et al. (2020) in our paper (we didn't add the one to Cadd et al. (2021) discussing climatic changes over the past 35 kyr as we focus on a period spanning from 75 to 35 ka). In our simulation, the water sources, water discharge and lake level are controlled by paleo-precipitation conditions inferred from paleo-climate simulation (Singarayer & Valdes, 2010), as a result our resistance maps (cost surfaces) vary through time and space. Because we are considering limited tectonic (subsidence and uplift) activity over the simulated time interval, and because of the temporal resolution of the model (e.g., we do consider mean annual precipitation regimes without accounting for seasonality or potential cyclonic regimes) our simulated drainage network remains relatively stable. Yet it is worth mentioning that our multiple flow direction algorithm prevents the locking of rivers into unique pathways and allows for a better representation of low-relief natural river system (such as branching river networks) (Salles et al., 2023).

REVIEWER COMMENTS

Reviewer #1 (Remarks to the Author):

I am happy with the changes and responses provided by authors and believe the manuscript is ready for publication.

Files associated with Reviewer #1's report:

<https://drive.google.com/drive/folders/19-dv2NlxIV4b45YmoNuKXbZg-2an9cw8?usp=sharing>

Reviewer #2 (Remarks to the Author):

The manuscript has improved. I'll address both the rebuttal and the text itself.

First, the authors make an incorrect claim in their rebuttal about the lack of use of demographic and cultural variables in recent models, especially in Crabtree et al 2021, and cited a specific sentence from that article (about "cultural influence") to try to reinforce that claim. I'm assuming that claim is based on an incomplete reading of the article by the authors. Crabtree et al factored in both demographic and ethnographic information to inform the kinds of travelers modeled as well as the kinds of decisions they would make as they moved. In particular, two of the three modeled scenarios (navigating based on visibility and navigating along rivers) rely directly on information provided by Aboriginal groups--and it worked. The river scenario is backed up by the findings of this manuscript, by the way, and it's worth noting that (especially with respect to extant ethnography). I think it is important for the authors to acknowledge the successful incorporation of demographic *and* ethnographic variables by other studies, in both their Introduction and Results section, the latter of which already has some text related to ethnography. It should also be noted that Crabtree et al specifically mention the complementarity of a localized foraging model framework, which I believe this paper provides to some degree. I recommend the authors emphasize the complementarity to that model instead of stating that their approach is an alternative. It's unnecessarily divisive. They do mention complementarity in some places, but it is inconsistent.

Second, I'm glad to see the removal of the simplistic least cost analysis. It wasn't helping their paper.

Third, modeling two entry points is definitely more than modeling just the original one, but is it sufficient? It's my understanding that there are more than two entry points that are discussed in the literature and hold analytical value. I will defer to the other reviewers in this area, but I will also point out that one of the likely reasons for why there are inconsistencies between the model used in this paper and the superhighways model of Crabtree et al is that the authors limited themselves to two entry points, whereas the superhighways model adopts a more holistic approach that uses the entire coastline of Sahul, all assumed-to-be-present water sources, and a dense regularly-spaced grid spanning the entire continent as possible origins. That is thousands upon thousands of origins compared to just two. There will of course be differences, which the authors should acknowledge. The fact that there is good alignment in many areas, despite only using two entry points, is very encouraging, and is worth highlighting. As I stated above, model complementarity should be a strength of this paper.

Fourth, there is still an issue with the "prospection" section of the paper. I understand that the authors are compute-limited in certain ways, which is something that they should explain. Right now the use of 5000 random points seems arbitrary and is definitely unexplained. They need to justify it. Why not use a regularly-spaced grid of points over the continent, which would ensure even sampling? With a random sample, you could end up with clusters of points in some areas and voids in others, which might be reflected in some of their observations about possible site locations. They should also say what they *would* do if they weren't compute-limited, i.e., calculate the map for every cell. It's

doable and opens the door to future work.

Last, but not least, I think there is a color scale issue with Figure 6 and Extended Data Figure 6. In both figures, if I am reading the text correctly, there is a graphic showing a heat map with both models combined, the only difference being a "clipping" of the landmass in the second figure to modern continental extents. I would expect to see the exactly the same colors in exactly the same places in both scenarios, i.e., the same values map to the same colors, but there are significant differences, making it difficult to compare them. If the graphics do in fact represent different things, that should be made more clear in the text and the figure captions.

Reviewer #3 (Remarks to the Author):

The authors have clearly taken the critical comments from all three reviewers on board and responded to these appropriately. There is evidence of significant and further work to make the model more robust and, where required (e.g. local site formation processes), conditional by adding appropriate caveats.

One section I made specific comments on needs some further corrections/amendments:

259 - 262 "Our results support the hypothesis of Hiscock and Wallis72 and other that Aboriginal people would have already settled in the Australian arid interior by the time more extreme conditions from the last glacial maximum (~20 ka)38,75 drastically changed those interior habitats with the expansion of new environmental barriers (e.g., sandy deserts).

On the basis of my comments and correct referencing this (more grammatical version) should read (or be a close variation of):

"Our results support the hypotheses of Hiscock and Wallis72 and Veth et al. 74 that Aboriginal people would have already settled in the Australian arid interior by the time more extreme conditions from the last glacial maximum (~20 ka)38, 41 transformed many of these interior habitats with the expansion of potential environmental barriers (e.g., sandy deserts)".

The Cadd et al. 2021 reference presents a fresh critique of LGM climatic and human responses but was rejected by the authors because it only examines systems after 35 ka; which post-dates their time tranche. However, it is precisely its relevance to new ways of understanding variability in LGM dynamics in Australia between 30 - 19 ka (such as colder and wetter conditions in the south east and not more arid, per se) that it should be included. This would be both for their discussion at this point of in situ desert adaptations as well as the important scholarship which brings climate data and proxies and the SahulArch database together for the first time.

Responses to reviewers' comments on the revised version of our manuscript.

We were very pleased with the outcomes of the review process of the revised manuscript, all reviewers were happy with the changes and responses that we made following their suggestions and comments: R1: *"I am happy with the changes and responses provided by authors and believe the manuscript is ready for publication"*; R2: *"The manuscript has improved"*; R3: *"The authors have clearly taken the critical comments from all three reviewers on board and responded to these appropriately"*.

In the following, we respond to the remaining comments. We answer suggestions of each reviewer and discussed the modifications that were made to address them.

REVIEWER 1

Comment: I am happy with the changes and responses provided by authors and believe the manuscript is ready for publication.

REVIEWER 2

Comment: The authors make an incorrect claim in their rebuttal about the lack of use of demographic and cultural variables in recent models, especially in Crabtree et al 2021, and cited a specific sentence from that article (about "cultural influence") to try to reinforce that claim. I'm assuming that claim is based on an incomplete reading of the article by the authors. Crabtree et al factored in both demographic and ethnographic information to inform the kinds of travellers modelled as well as the kinds of decisions they would make as they moved. In particular, two of the three modelled scenarios (navigating based on visibility and navigating along rivers) rely directly on information provided by Aboriginal groups--and it worked. The river scenario is backed up by the findings of this manuscript, by the way, and it's worth noting that (especially with respect to extant ethnography). I think it is important for the authors to acknowledge the successful incorporation of demographic *and* ethnographic variables by other studies.

Our understanding of the model proposed by Crabtree et al. 2021 is that the pedestrian-transportation networks is based on a FETE approach (*from everywhere to everywhere*) to generate multiple least-cost routes based on elevation data, land cover, and physical traveller characteristics. In their study, both the elevation and land cover are considered static. As for the physical traveller characteristics, their proposed simulations consider a typical 25-year-old woman and travellers move across Sahul while minimising caloric costs.

We did not find in the paper any reference regarding how these travellers' movements were inferred by information provided by Aboriginal groups. In the Methods section of the Crabtree et al. paper, it is stated that the FETE's anisotropic cost-minimization function relies on well-established caloric expenditure with references (85 to 89 are provided below) that do not seem to be specifically related to Aboriginal groups.

85. Pandolf, K. B., Givoni, B. & Goldman, R. F. Predicting energy expenditure with loads while standing or walking very slowly. *J. Appl. Physiol.* 43, 577–581 (1977).

86. Looney, D. P. et al. Metabolic costs of military load carriage over complex terrain. *Mil. Med.* 183, e357–e362 (2018).

87. Soule, R. G. & Goldman, R. F. Terrain coefficients for energy cost prediction. *J. Appl. Physiol.* 32, 706–708 (1972).

88. Santee, W., Blanchard, L., Speckman, K., Gonzalez, J. & Wallace, R. Load Carriage Model Development and Testing with Field Data (USARIEM, 2003).

89. Wood, B. M. & Wood, Z. J. Energetically optimal travel across terrain: visualizations and a new metric of geographic distance with anthropological applications. in *Proc. SPIE 6060 Visualization and Data Analysis 2006* (eds. Erbacher, R. F. et al.) 60600F-1–60600F-7 (2006).

As pointed out in our first *"responses to reviewers"* document, Crabtree et al. 2021 study does not include *demographic composition, traveller velocity, the types of eco-systems encountered and cultural influence*. Similarly, Bradshaw et al. 2023

simulations *consider ecological parameters alone* while acknowledging that social, cultural, and economic decisions would have also played a role on the timing and spread of human dispersal in Sahul.

It is worth mentioning that our approach accounts for reported variability ethnographic data from Australian Aboriginal people (Kelly, 2007 – as discussed in the first section of our Results) as well as differences between residential moves across Aboriginal groups (Anbarra people from Arnhem Land and Ngadadjara foragers in the Australian western desert). In addition, and as already pointed out in the first “responses to reviewers” document, the Lévy walker approach has been shown to accurately represent contemporary hunter-gatherer populations (discussed in the second paragraph of the introduction and the conclusion), populations whose behaviours and decision-making processes are embedded with cultural knowledge and practices.

Comment: It should be noted that Crabtree et al specifically mention the complementarity of a localized foraging model framework, which I believe this paper provides to some degree. I recommend the authors emphasize the complementarity to that model instead of stating that their approach is an alternative. It's unnecessarily divisive. They do mention complementarity in some places, but it is inconsistent.

We agree with the reviewer’s comment regarding the complementarity of the approaches and have added the following sentence in the second section of the Discussion:

As a result, the proposed Lévy walk foraging framework complements well existing pedestrian-transportation networks used to evaluate the timing and spread of human dispersal in Sahul^{2,9}.

Comment: Modelling two entry points is definitely more than modelling just the original one, but is it sufficient? It's my understanding that there are more than two entry points that are discussed in the literature and hold analytical value. I will defer to the other reviewers in this area, but I will also point out that one of the likely reasons for why there are inconsistencies between the model used in this paper and the superhighways model of Crabtree et al is that the authors limited themselves to two entry points, whereas the superhighways model adopts a more holistic approach that uses the entire coastline of Sahul, all assumed-to-be-present water sources, and a dense regularly-spaced grid spanning the entire continent as possible origins. That is thousands upon thousands of origins compared to just two. There will of course be differences, which the authors should acknowledge. The fact that there is good alignment in many areas, despite only using two entry points, is very encouraging, and is worth highlighting. As I stated above, model complementarity should be a strength of this paper.

Our understanding of Crabtree et al. 2021 model is that the *origin* locations for travel are not specifically located on a couple of entry points, nor are they solely distributed over the entire coastline of Sahul. Instead, depending on the tested hypotheses, either a regularly spaced grid with spacing selected as one point every 50 surface cells in each direction or the shoreline cells combined with Water Observations from Space dataset (including rivers and lakes) were used for modelling travel and optimal superhighways. In their case, the least-cost models of pedestrian movement do include points all over the continents and do not look directly at peopling from possible entry points. Latest work from Bradshaw et al. 2023 combine the results from the superhighways model with a stochastic-ecological, cellular-automaton model in which they force entry points to evaluate the initial spread and timing of human dispersal across Sahul.

Here assuming a northern and southern route, we follow a similar approach to the ones in Bradshaw et al. 2021 and Bradshaw et al. 2023. As pointed out in our paper (last section of the Results) *peopling of Sahul was likely accomplished in successive waves, unrestricted to the chosen northern and southern entries^{2,8,10,27} nor to a 75-73 ka double-entry scenario²*. It is worth mentioning that the superhighways in Crabtree et al. 2021 are well-documented ethnographic trade routes that are younger than the period we model (initial peopling). In their model, the landscape remains intact over the simulation. It is possible that incorporating changes related to geomorphological evolution would modify some of the primary corridors they have identified.

Comment: There is still an issue with the "prospection" section of the paper. I understand that the authors are compute-limited in certain ways, which is something that they should explain. Right now, the use of 5000 random points seems arbitrary. They need to justify it. Why not use a regularly-spaced grid of points over the continent, which would ensure even sampling? With a random sample, you could end up with clusters of points in some areas and voids in others, which

might be reflected in some of their observations about possible site locations. They should also say what they *would* do if they weren't compute-limited, i.e., calculate the map for every cell. It's doable and opens the door to future work.

In our study, our probabilistic assessment produces 100 billion paths (5,000 realisations over 10 million steps each for each entry point) comparable to the number of migratory pathways computed in Crabtree et al. 2021 (i.e., 125 billion). From the Lévy walks, we can aggregate the 100 billion paths to produce heat maps of most likely walkers' distribution as shown in Fig. 6 and Extended Data Fig. 6. The computation required to produce those maps is equivalent the one performed in the superhighways model. However, the evaluation of residential likelihood similar to what we present for the archaeological sites and in the prospective section of the paper is much more computational demanding. It involves assessing each realisation of 10 million steps individually (Extended Data Fig. 4 & 5) for each site to extract all the walkers close to the site and then to group them (k-means clustering) based on the calculated cumulative travelled distance to the initial entry point (see Results first section) to estimate the number of distinct visits.

Our random selection of 5,000 points over the Australian continent in the prospection section is given on the above figure and we have added Extended Data Fig. 7 that shows this distribution. In the caption of the figure, we also explain the calculations related to the likelihood evaluation. We first calculate travelled distances for each of these points and produce a map of mean walked distances for Australia. We then evaluate amongst these 5,000 hypothetical sites the ones that are most likely to be occupied based on the number of predicted visits and the number of walkers in their surroundings. Both calculation steps are performed on the computed paths from the Lévy-walk mechanistic simulations. As pointed by the reviewer, the residential likelihood is site-specific and changing site locations might change the selected 10 most likely occupied sites. In our view this section is used to showcase one of the interesting outcomes of the proposed approach. We have modified the last part of the section to clearly state that because the resolution of the simulation is at 2 km, we envision that this method might be better suited to look at a specific (more constrained) region of interest prior to an archaeological campaign rather than the entire continent.

Comment: I think there is a colour scale issue with Figure 6 and Extended Data Figure 6. In both figures, if I am reading the text correctly, there is a graphic showing a heat map with both models combined, the only difference being a "clipping" of the landmass in the second figure to modern continental extents. I would expect to see the same colours in exactly the same places in both scenarios, i.e., the same values map to the same colours, but there are significant differences, making it difficult to compare them. If the graphics do in fact represent different things, that should be made clearer in the text and the figure captions.

We thank the reviewer for picking the colour scale issue between Fig. 6 and Extended Data Fig. 6 and have updated both figures to have the same colour range in the 2 scenarios.

REVIEWER 3

Comment: One section I made specific comments on needs some further corrections/amendments:

259 - 262 *"Our results support the hypothesis of Hiscock and Wallis⁷² and other that Aboriginal people would have already settled in the Australian arid interior by the time more extreme conditions from the last glacial maximum (~20 ka) drastically changed those interior habitats with the expansion of new environmental barriers (e.g., sandy deserts)."*

On the basis of my comments and correct referencing this (more grammatical version) should read (or be a close variation of): *"Our results support the hypotheses of Hiscock and Wallis⁷² and Veth et al. ⁷⁴ that Aboriginal people would have already settled in the Australian arid interior by the time more extreme conditions from the last glacial maximum (~20 ka) transformed many of these interior habitats with the expansion of potential environmental barriers (e.g., sandy deserts)".*

Following reviewer's suggestion, we have modified the last sentence of this paragraph accordingly:

Our results support the hypotheses of Hiscock and Wallis⁶⁸ and Veth et al.⁶⁹ that Aboriginal people would have already settled in the Australian arid interior by the time more extreme conditions over the last glacial maximum (35 to 15 ka)^{34,37,72} transformed many of these interior habitats with the expansion of potential environmental barriers (e.g., sandy deserts)^{65,70,71} and variable biological productivity⁷².

Comment: The Cadd et al. 2021 reference presents a fresh critique of LGM climatic and human responses but was rejected by the authors because it only examines systems after 35 ka, which post-dates their time tranche. However, it is precisely its relevance to new ways of understanding variability in LGM dynamics in Australia between 30 - 19 ka (such as colder and wetter conditions in the southeast and not more arid, per se) that it should be included.

Based reviewer's comment, we have added the reference to Cadd et al. 2021 in the discussion first section *Predicted peopling of Sahul* (end of the second paragraph). Reference 72 in the blue sentence above.

Reviewers' Comments:

Reviewer #2:

Remarks to the Author:

I need to address two comments in the latest rebuttal provided by the authors.

1) "We did not find in the paper any reference regarding how these travellers' movements were inferred by information provided by Aboriginal groups. In the Methods section of the Crabtree et al. paper, it is stated that the FETE's anisotropic cost-minimization function relies on well-established caloric expenditure with references that do not seem to be specifically related to Aboriginal groups."

Okay, let's try this one more time, since the first two times I asked you to read that paper more carefully, you clearly did not do so. Focusing on the methods section of the paper was not the right thing to do. Look at the methods summary in the main text. The traveler's physical characteristics and walking patterns are modeled on ethnographic data associated with Aboriginal women. The preference for visually prominent landscape features comes directly from interviews with Aboriginal groups. I could go on, but those two examples make my point. Please stop misreading the paper and reference it appropriately in your manuscript with respect to the use of demographic and cultural variables.

2) "Our understanding of Crabtree et al. 2021 model is that the origin locations for travel are not specifically located on a couple of entry points, nor are they solely distributed over the entire coastline of Sahul. Instead, depending on the tested hypotheses, either a regularly spaced grid with spacing selected as one point every 50 surface cells in each direction or the shoreline cells combined with Water Observations from Space dataset (including rivers and lakes) were used for modelling travel and optimal superhighways. In their case, the least-cost models of pedestrian movement do include points all over the continents and do not look directly at peopling from possible entry points....It is worth mentioning that the superhighways in Crabtree et al. 2021 are well-documented ethnographic trade routes that are younger than the period we model (initial peopling)."

Your understanding is incorrect. If you would like to inspect the 94,332 distinct shoreline travel origin points used in that study for yourselves, here is a link to the shapefile that contains them: https://www.dropbox.com/scl/fi/ntl84y1ztwzrcscu2fkin/sahul_coastline_points.zip?rlkey=nevat5emourkaebajkm36hlik&dl=0. Several of the modeled scenarios use the coastline as the starting (entry) point for travel to water sources across the entire continent, a regular grid that spans the entire continent, and the entire coastline itself. See Table 1 in that paper (hint: it's in the main text, not the methods section). That more than covers the initial peopling scenario—especially when they are combined...which is what the authors did. The comment about "well-documented trade routes" demonstrates a complete lack of understanding of that approach and is just plain disingenuous. While it's very encouraging and very interesting to know that routes associated with movement during initial peopling (it's in the title of that paper!) might align well with how people moved in later time periods, that was not the focus of the work and is not relevant to this discussion. Please adjust your manuscript per my last set of comments on this topic.

Responses to reviewer' comments.

In our previous answer to reviewer's comments, we genuinely tried to answer to the points that were made, but we feel there might have been some misunderstanding that we hope to have clarified in this revision.

REVIEWER 2

Comment: Okay, let's try this one more time, since the first two times I asked you to read that paper more carefully, you clearly did not do so. Focusing on the methods section of the paper was not the right thing to do. Look at the methods summary in the main text. The traveler's physical characteristics and walking patterns are modeled on ethnographic data associated with Aboriginal women. The preference for visually prominent landscape features comes directly from interviews with Aboriginal groups. I could go on, but those two examples make my point. Please stop misreading the paper and reference it appropriately in your manuscript with respect to the use of demographic and cultural variables.

Following reviewer's comment, we have modified the text to acknowledge the incorporation of demographic and ethnographic variables in other studies/models. To do so we have removed the following line in the second paragraph of the discussion section on **Foraging and migrating across transient landscapes**:

"Previous approaches^{2,4,8,9} predict peopling in terms of settlement patterns without considering the role local foraging might have played in the decision process that entails the choice of a particular migration route."

We have also added the following in the same section:

"As a result, the proposed Lévy walk foraging framework complements well existing pedestrian-transportation network models which account for ethnographic and demographic data^{2,7,9} and that have been previously used to evaluate the timing and spread of human dispersal in Sahul^{2,9}."

In the conclusion, we made the following changes based on reviewer's comment:

"As pointed out for other models^{2,4,8,9}; Our method does not account for social, cultural and economic factors;... it"

Comment: Your understanding is incorrect. If you would like to inspect the 94,332 distinct shoreline travel origin points used in that study for yourselves, here is a link to the shapefile that contains them: dropbox. Several of the modeled scenarios use the coastline as the starting (entry) point for travel to water sources across the entire continent, a regular grid that spans the entire continent, and the entire coastline itself. See Table 1 in that paper (hint: it's in the main text, not the methods section). That more than covers the initial peopling scenario—especially when they are combined...which is what the authors did. The comment about "well-documented trade routes" demonstrates a complete lack of understanding of that approach and is just plain disingenuous. While it's very encouraging and very interesting to know that routes associated with movement during initial peopling (it's in the title of that paper!) might align well with how people moved in later time periods, that was not the focus of the work and is not relevant to this discussion. Please adjust your manuscript per my last set of comments on this topic.

Following reviewer's comment, we have modified the manuscript to reflect the complementarity of the double-entry scenario and the differences with the approach proposed in Crabtree et al. (2021).

Also based on the reviewer's comment on the previous revision, we are now highlighting the fact that there is good alignment in many areas, despite only using two entry points.

In the second paragraph of the discussion section on **Predicted peopling of Sahul**:

"We find that probability of walkers' presence is high in the vicinity of the major corridors⁹ (Fig. 6)."

and

"In addition, those differences might be induced by the choice of a double-entry scenario² where the superhighways model adopts a more holistic approach that uses different sets as origins or destinations⁹ (e.g., grid-to-grid, grid-to-coastline and grid-to-water). The fact that there is good alignment in many areas, despite only using two entry points, is very encouraging and suggests that our approach can reproduce some of the major migration routes proposed in the superhighways model²."

Last paragraph of the result section:

“Second, peopling of Sahul was likely accomplished in successive waves, unrestricted to the chosen northern and southern entries^{2,8,10,27} nor to a 75-73 ka double-entry scenario².”

We have also added the following in the last sentence to emphasize the complementarity to existing models for the region as suggested by the reviewer in his previous review:

*“We believe our approach **complements existing models^{2,4,8,9} and offers...***”

Reviewers' Comments:

Reviewer #2:

Remarks to the Author:

No additional comments.